# Quantum transport evidence of isolated topological nodal-line fermions

Hoil Kim[1,2], Jong Mok Ok[1,2,3], Seyeong Cha [4], Bo Gyu Jang [5],
Chang Il Kwon [1,2], Yoshimitsu Kohama [6], Koichi Kindo[6], Won Joon Cho[7],
Eun Sang Choi[8], Youn Jung Jo[9], Woun Kang [10], Ji Hoon Shim [5],
Keun Su Kim [4] & Jun Sung Kim [1,2] ✉

Anomalous transport responses, dictated by the nontrivial band topology, are the key for application of topological materials to advanced electronics and spintronics. One promising platform is topological nodal-line semimetals due to their rich topology and exotic physical properties. However, their transport signatures have often been masked by the complexity in band crossings or the coexisting topologically trivial states. Here we show that, in slightly hole-doped $SrAs_3$, the single-loop nodal-line states are well-isolated from the trivial states and entirely determine the transport responses. The characteristic torus-shaped Fermi surface and the associated encircling Berry flux of nodal-line fermions are clearly manifested by quantum oscillations of the magnetotransport properties and the quantum interference effect resulting in the two-dimensional behaviors of weak antilocalization. These unique quantum transport signatures make the isolated nodal-line fermions in $SrAs_3$ desirable for novel devices based on their topological charge and spin transport.

Topological semimetals[1–4], a class of quantum states with symmetry-protected band crossings, have attracted tremendous interest recently because of their nontrivial topology, the presence of the peculiar surface states, and the resultant exotic electromagnetic responses[5,6]. Among many types of topological semimetals, nodal-line semimetals (NLSMs) arguably offer the most fascinating quantum system with rich topological structures[7–9] and electronic correlations[10,11]. In NLSMs, the crossings of conduction and valence bands extend along one-dimensional lines in the momentum space, which can have various topologically distinct forms, e.g., an extended line across the entire Brillouin zone (BZ), a single closed loop inside the BZ, or a chain of multiple loops knotted or linked together[12–14]. In real systems with a finite carrier density, these nodal lines are enclosed by a thin tubular

Fermi surface (FS), on which the associated $\pi$ Berry flux imprints the characteristic smoke-ring-shaped pseudospin texture[15–17]. These unique topological characteristics of nodal-line fermions are expected to induce exotic charge and spin transport phenomena such as electric-field-induced anomalous transverse current[17,18], large weak antilocalization[19], spin-polarized filtering[20], and anomalous Andreev reflection[21,22], most of which are yet to be realized in experiments.

One major obstacle to investigating the unique transport phenomena of nodal-line fermions is the lack of suitable materials. Thus far, experimental studies on NLSMs have focused on the verification of nodal-line electronic structures, using angle-resolved photoemission spectroscopy (ARPES)[23–26] or de Haas-van Alphen (dHvA) oscillations[27–30], not on their unique transport properties. This is

[1]Center for Artificial Low Dimensional Electronic Systems, Institute for Basic Science (IBS), Pohang 37673, Korea. [2]Department of Physics, Pohang University of Science and Technology (POSTECH), Pohang 37673, Korea. [3]Department of Physics, Pusan National University, Busan 46241, Korea. [4]Department of Physics, Yonsei University, Seoul 03722, Korea. [5]Department of Chemistry, Pohang University of Science and Technology (POSTECH), Pohang 37673, Korea. [6]Institute for Solid State Physics, University of Tokyo, Kashiwa, Chiba 277-8581, Japan. [7]Material Research Center, Samsung Advanced Institute of Technology (SAIT), Samsung Electronics Co., Ltd, Suwon-si, Gyeonggi-do 16678, Korea. [8]National High Magnetic Field Laboratory, Florida State University, Tallahassee, FL 32310, USA. [9]Department of Physics, Kyungpook National University, Daegu, Korea. [10]Department of Physics, Ewha Womans University, Seoul 03760, Republic of Korea. ✉e-mail: js.kim@postech.ac.kr

because most NLSM candidates possess complex multiple nodal loops linked together[23–25,28]; only a handful of candidates[31,32] are expected to have a single loop and the corresponding torus-shaped FS in the BZ. Furthermore, in many cases, there exist topologically trivial states at the Fermi level ($E_F$) that provide additional conduction channels, which hampers the identification of the characteristic transport properties of nodal-line fermions alone. Here, we present an NLSM candidate SrAs$_3$ as a model system in which the quantum transport responses are entirely dictated by nodal-line fermions from a single torus-shaped FS without other trivial states. Shubnikov-de Hass (SdH) oscillations confirm dominant charge conduction by nodal-line fermions in slightly hole-doped SrAs$_3$ and identify its tubular FS, thinnest among those of known NLSMs, and the characteristic smoke-ring-type pseudospin texture. These unique characters of nodal-line fermions are further corroborated by the quantum interference effect with disorder-induced scattering, resulting in unusual two-dimensional behaviors of weak antilocalization (WAL) and its strong variation to the FS characters.

## Results

### Nodal-line electronic structure of SrAs$_3$

SrAs$_3$, a member of the material class $AEPn_3$ ($AE$ = Ca, Sr, Ba, and $Pn$ = P, As)[33–35], has been suggested as a promising candidate NLSM[32]. It consists of buckled As planes, sandwiched by Sr atoms and staked along the $c$-axis in a monoclinic structure (space group $C2/m$) (Fig. 1a). Band crossings accidentally occur between the conduction and valence bands, derived from the $p$ orbital states of two inequivalent As sites[32] (Supplementary Fig. 1). The resultant single nodal-loop is expected to be located around the Y point in the BZ on the $a$–$c$ plane, or ($k_x$,$k_y$) plane in the momentum space, where $k_x$, $k_y$ and $k_z$ denote the orthogonal basis vectors parallel to the crystal axis $a$, the reciprocal lattice vector $k_c$, and the crystal axis $b$, respectively (Fig. 1d). This nodal-loop structure at low-energy states has been suggested by band calculations[32] and ARPES[36] and partly by quantum oscillations[37,38]. Our ARPES intensity plots in a wide energy region along the $k_x$ and $k_y$ directions clearly show that the low-energy states are only located near the Y point (Fig. 1e, h), in good agreement with the band structure

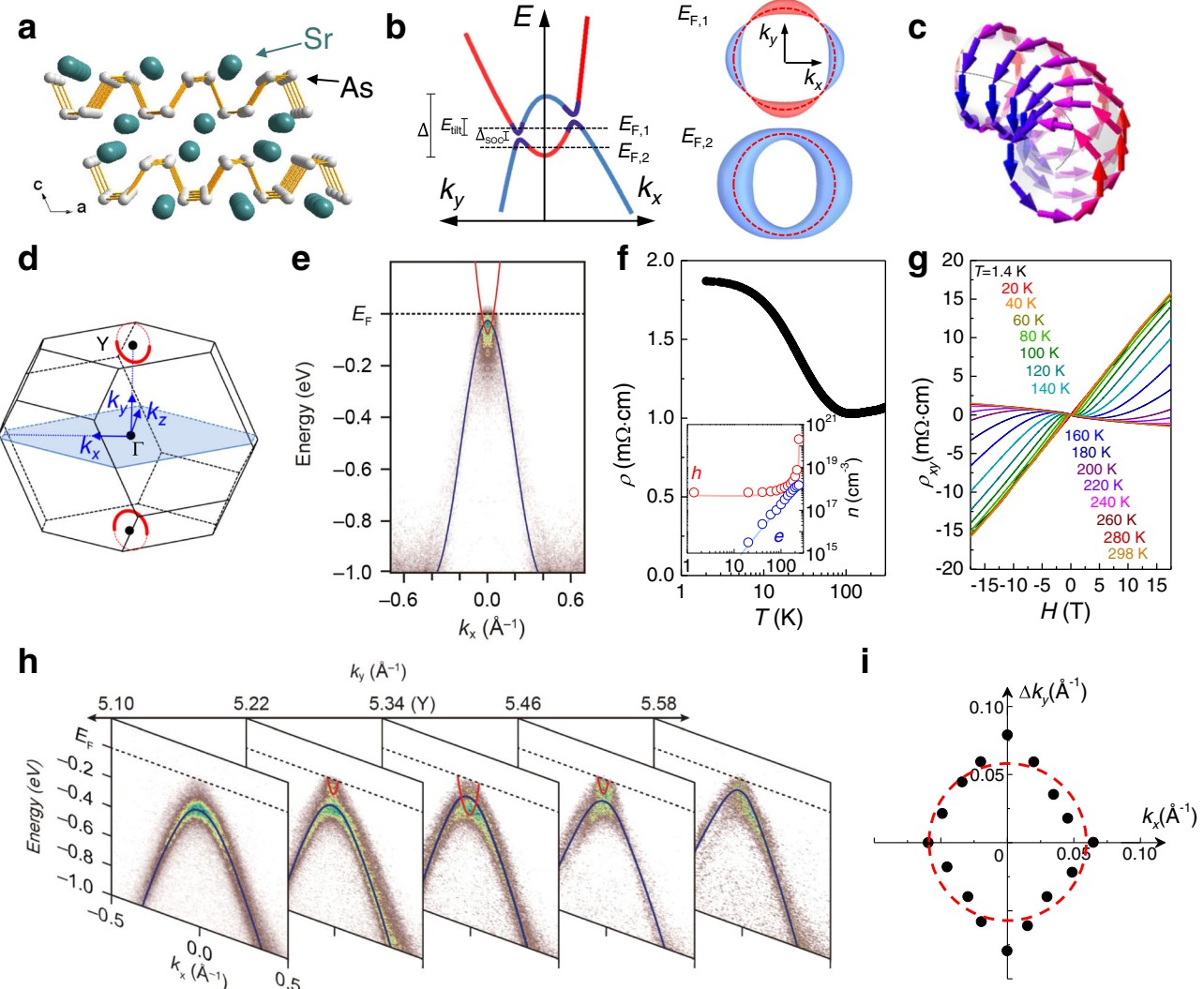

**Fig. 1 | Crystal and electronic structures of a nodal-line semimetal SrAs$_3$. a** The crystal structure of SrAs$_3$. **b** The schematic band crossing for asymmetric nodal-line states with a tilted energy dispersion ($E_{tilt}$), a finite spin−orbit-coupling gap ($\Delta_{SOC}$) and a band overlap energy ($\Delta$). The corresponding Fermi surfaces at different Fermi levels ($E_F$) are shown in the right, a crescent-type for $E_{F,1}$ and a torus-type for $E_{F,2}$. **c** The smoke-ring-type pseudospin texture imprinted on the Fermi surface. **d** The Brillouin zone of SrAs$_3$ with a single nodal ring (red circle) centered at the Y point. **e** The ARPES spectra of SrAs$_3$ taken at the Y point along $k_x$ with the photon energy of 99 eV. The overlaid red and blue lines indicate the conduction and valence bands, respectively. **f** The temperature dependence of the in-plane resistivity ($\rho$). The inset shows the carrier densities ($n$) for electron ($e$) and hole ($h$). **g** The magnetic field-dependent Hall resistivity ($\rho_{xy}$) of SrAs$_3$ at different temperatures. **h** A series of ARPES spectra taken along $k_x$ at different photon energies (85-104 eV) corresponding to $k_y$ marked on top of each panel. **i** The nodal-ring of the crossing points between the conduction and valence bands in ARPES data, with dashed red circle as a guide to the eye.

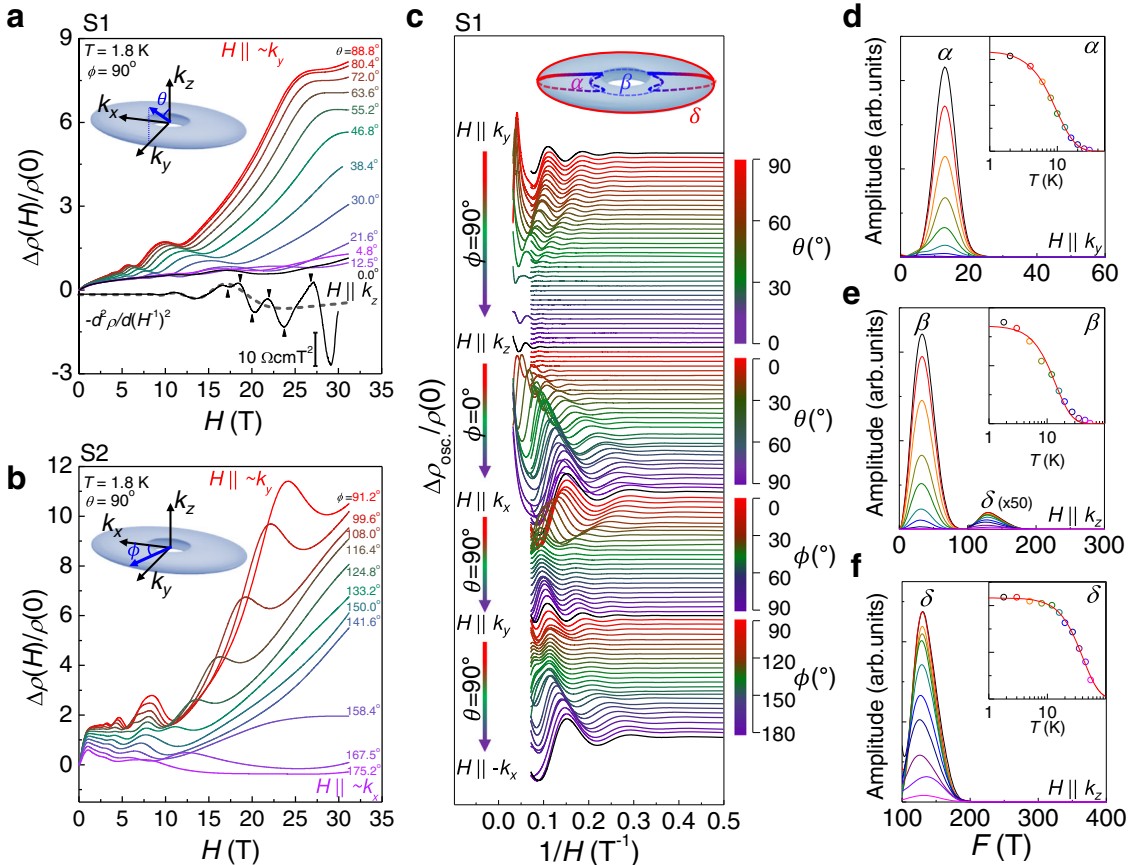

**Fig. 2 | Shubnikov-de Hass oscillations of SrAs₃. a, b** Magnetoresistance (MR) $\Delta\rho(H)/\rho(0)$ of SrAs$_3$ with different magnetic field orientations in the $(k_y, k_z)$ plane (**a**, S1) and in the $(k_x, k_y)$ plane (**b**, S2). The second derivative of $\rho(H)$ with respect to $1/H$ at $H\|k_z$ is also shown in **a**. The overlaid dashed gray curve corresponds to the coexisting SdH oscillations with a lower frequency. The black arrows indicate peaks and deeps of the higher-frequency oscillations. The polar ($\theta$) and azimuthal ($\phi$) angles are defined with respect to the torus-shaped Fermi surface as shown in the insets. **c** SdH oscillations ($\Delta\rho_{osc}/\rho(0)$) at various magnetic field orientations in the planes of $(k_x, k_y)$, $(k_y, k_z)$ and $(k_x, k_z)$ for S1. The inset shows torus-shaped Fermi surface of SrAs$_3$ with the poloidal orbit ($\alpha$) and the inner ($\beta$) and outer ($\delta$) toroidal orbits. **d, e, f** Fast Fourier transform (FFT) amplitudes for $\alpha$ orbit (**d**), for $\beta$ orbit (**d**) and $\delta$ orbit (**f**), taken at various temperatures for $H\|k_y$ (**d**) and $H\|k_z$ (**e, f**). The insets show the temperature-dependent FFT amplitudes, together with the fits (red lines) to the Lifshitz–Kosevich equation. In **e**, the FFT amplitude of the $\delta$ orbit, $F > 100$ T, is magnified for comparison.

calculations using the modified Becke–Johnson exchange potential (Supplementary Fig. 3). Since the nodal-loop is expected to be located in the $(k_x, k_y)$ plane centered at the Y point, a series of ARPES data taken along the $k_x$ axis was collected at different $k_y$'s across the Y point (Fig. 1h). The evolution of ARPES spectra along the $k_y$ direction clearly reveals that band crossings only occur near the Y point of the BZ. The radius of the nodal-loop $K_0$ is estimated to be -0.057 Å$^{-1}$ (Fig. 1i), consistent with the results of quantum oscillations, discussed below.

Having established the nodal-line electronic structure in SrAs$_3$, we now focus on the details of the FS, which cannot be directly resolved by APRES due to its small size. In SrAs$_3$, unlike the ideal nodal-loop, the conduction and valence bands have asymmetric dispersion, which introduces a tilting of the nodal-loop with a characteristic energy scale $E_{tilt}$, smaller than the band overlap energy $\Delta$ (Fig. 1b). Furthermore, the finite spin–orbit-coupling (SOC) lifts the band degeneracy at the nodal-loop and induces a small momentum-dependent SOC gap $\Delta_{SOC}$. Thus, the torus-shaped FS is only established when $\varepsilon_F$, the energy difference between $E_F$ and the band crossing point in the momentum-energy space, is larger than $\Delta_{SOC}/2$ and $E_{tilt}$ but smaller than $\Delta/2$. This characteristic torus-shaped FS possesses the smoke-ring-type pseudospin texture (Fig. 1c) and the associated $\pi$ Berry flux, which disappears, e.g., in a drum-shaped FS for $\varepsilon_F > \Delta/2$. Therefore, proper adjustment of $\varepsilon_F$ is needed to access the unique transport properties of nodal-line fermions in SrAs$_3$. We carefully selected the crystals that showed a dominant carrier type at low temperatures, using the in-plane

resistivity, $\rho$, (Fig. 1f) and Hall resistivity, $\rho_{xy}$ (Fig. 1g). At high temperatures, all crystals exhibit the nonlinear field dependence of $\rho_{xy}(H)$, which originates from the two-band conduction of thermally excited electrons and holes, as commonly observed in many topological semimetals with low carrier densities[39,40]. Using the two-band conduction model, we estimated the temperature dependence of electron ($n_e$) and hole ($n_h$) carrier densities. Some samples show that the electron density $n_e$ is drastically reduced at low temperatures, smaller by one or two orders of magnitude than the hole density $n_h = 3-7 \times 10^{17}$ cm$^{-3}$ (Supplementary Fig. 4 and Table 1). These samples are used to measure both SdH oscillations and quantum interference effect for $H\|k_y$, while two representative samples with a relatively large hole carrier density (S1 and S2) were used for investigating full angle-dependent SdH oscillations.

**Torus-shaped Fermi surface**

The magnetoresistance (MR), $\Delta\rho(H)/\rho(0)$, taken at high-magnetic fields up to 31.6 T for various field directions, is presented for the selected crystals (S1 and S2) in Fig. 2a, b. As compared to dHvA oscillations, SdH oscillations in the MR directly access the FSs responsible for charge conduction. For a torus-shaped FS (the inset of Fig. 2c), a small cyclotron orbit ($\alpha$) on the poloidal plane is expected under magnetic fields in the nodal-loop plane, here $H\|$ $(k_x, k_y)$ plane. For the magnetic field normal to the nodal-loop plane, $H\|k_z$, two extremal inner ($\beta$) and outer ($\delta$) toroidal orbits, significantly different in size, are

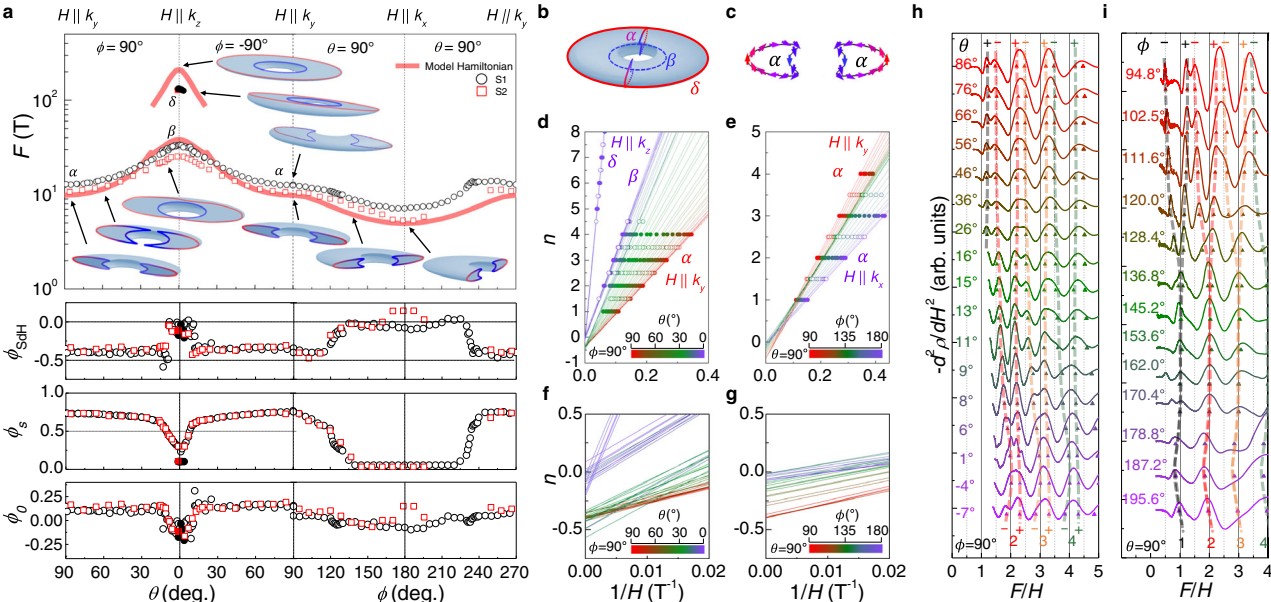

**Fig. 3 | Toroidal Fermi surface and Berry phase evolution of SrAs₃. a** Angle-dependent SdH frequency ($F$) and the phase offset of SdH oscillation ($\phi_{SdH}$) for two samples S1 (black) and S2 (red). The spin-splitting phase ($\phi_s$) and the characteristic phase ($\phi_0$) are also shown in the lower panels. The calculated $F$ using the model Hamiltonian is overlaid with red lines. The corresponding extremal orbits on the torus-shaped Fermi surface are also presented for selected field orientations in the inset. **b** Torus-shaped Fermi surface of SrAs₃ with the poloidal orbit ($\alpha$) and the inner ($\beta$) and outer ($\delta$) toroidal orbits. **c** Poloidal cross-section of the Fermi surface ($\alpha$) with pseudospin textures indicated by the arrows. **d–g** Landau fan diagram for various field orientations with different polar ($\theta$) angles (**d**, **f**) and azimuthal ($\phi$) angles (**e**, **g**) for S1. The maxima (solid circles) and minima (open circles) of $\Delta\rho(H)/\rho(0)$ are assigned with integer and half-integer of the Landau index. **h, i** The second derivative of $\rho(H)$, $-d^2\rho/dH^2$, as a function of the nomalized $F/H$ for various magnetic field orientations with different polar ($\theta$) (**h**) and azimuthal ($\phi$) angles (**i**) for S2. The spin-splitting peaks of SdH oscillations are indicated by triangle symbols. The shaded dashed lines correspond to the spin-split Landau levels, indicated by the color-coded integer index and the + and − symbols.

expected[41]. These characteristic behaviors of SdH oscillations are indeed observed in experiments (Fig. 2a, b). As the magnetic field orientation changes in the ($k_y,k_z$) plane, SdH oscillations with a small frequency $F$ vary systematically with the polar angle ($\theta$) as the cyclotron orbit changes from $\alpha$ to $\beta$. Near $H\|k_z$, additional oscillations with a high $F$ ~ 129 T are detected, which is more clearly visible in the second derivative curve of $\rho(H)$, $-d^2\rho/d(H^{-1})^2$ (Fig. 2a). This additional cyclotron orbit with a large size corresponds to the outer toroidal orbit ($\delta$). For $H\|$ ($k_x,k_y$) plane (Fig. 2b), SdH oscillations are well described by a single SdH frequency, corresponding to the poloidal orbit ($\alpha$). The SdH oscillations with a single frequency $F$ are described by the Lifshitz–Kosevich (LK) formula[42–44],

$$\Delta\sigma_{xx} \propto R_T R_D \left[\cos 2\pi\left(\frac{F}{H}+\phi_0+\frac{\phi_s}{2}\right) + \cos 2\pi\left(\frac{F}{H}+\phi_0-\frac{\phi_s}{2}\right)\right], \quad (1)$$

where $R_T$ and $R_D$ are damping factors due to a finite temperature and scattering, respectively. The characteristic phase $\phi_0$ and spin-splitting phase $\phi_s$ are two major components determining the phase offset of SdH oscillations $\phi_{SdH}$, as discussed below. From the temperature-dependent SdH oscillations, we estimate the cyclotron effective mass of each orbit, yielding $m^*/m_e = 0.076(5)$, 0.23(1), and 0.079(3) for the $\alpha$, $\beta$, and $\delta$ orbits, respectively (Fig. 2d, e, and f). The estimated quantum scattering times from the field-dependent SdH oscillations are $\tau_q = 0.085(8)$, 0.075(7), and 0.010(1) ps for the $\alpha$, $\beta$, and $\delta$ orbits, respectively, which are relatively long, as typically observed in topological semimetals.

A small deviation from the ideal torus-shaped FS is well resolved in the detailed angle dependence of the SdH frequency (Fig. 3a). To this end, we constructed a general two-band model Hamiltonian near the Y point $H(\mathbf{k}) = \sum_{i=0}^{3} g_i(\mathbf{k})\sigma_i$, where $\sigma_0$ is the identity matrix, $\sigma_{1,2,3}$ are Pauli matrices, and $g_i(\mathbf{k})$ is the real function of $\mathbf{k}$.[32] Considering three symmetries at Y point, time-reversal symmetry $\hat{T} = K$ with spinless

complex conjugate operator $K$, inversion symmetry $\hat{P} = \sigma_z$, and mirror symmetry $\hat{M}: k_x \leftrightarrow k_x; k_y \leftrightarrow k_y; k_z \leftrightarrow -k_z$, the coefficients $g_i(\mathbf{k})$ for the lowest orders of $\mathbf{k}$ are described by $g_0(\mathbf{k}) = a_0 + a_1 k_x^2 + a_2 k_y^2 + a_3 k_z^2$, $g_2(\mathbf{k}) = b_3 k_z$, and $g_3(\mathbf{k}) = m_0 + m_1 k_x^2 + m_2 k_y^2 + m_3 k_z^2$. The parameters $a_i$, $b_3$, $m_i$ are obtained to match the calculated cross-sectional size of FS with the measured SdH frequency as a function of the polar ($\theta$) and azimuthal ($\phi$) angles (Fig. 3a and Supplementary Table 2). Unlike the ideal torus-shaped FS, the resultant FS of SrAs₃ has a crescent-shaped poloidal cross-section, rather than the circular one and exhibits a small momentum-dependent asymmetry within the nodal plane. Along the toroidal direction, a finite tilting energy $\Delta_{tilt}$ ~ 5 meV, far smaller than the band overlap energy $\Delta$ ~ 120 meV and the Fermi level $E_F$ ~ 50 meV, introduces $\phi$-dependent distortion, leading to a weak variation of the SdH frequency. A detailed comparison between model calculations and experiments is provided in the Supplementary Note 4. The volume of FS and the corresponding carrier density $n_h$ ~ $1.7 \times 10^{18}$ cm⁻³ are in reasonable agreement with $n_h$ ~ $7 \times 10^{17}$ cm⁻³ from the Hall effect. The radius of the nodal-loop $K_0$ ~ 0.065 Å⁻¹, estimated from the constructed FS, agrees well with the APRES results (Fig. 1i). In addition, the calculated cyclotron masses using the constructed Hamiltonian are consistent with the experimental values for $H\|k_x$, $k_y$, and $k_z$ (Supplementary Table 2). Moreover, the band overlap energy $\Delta$ ~ 120 meV from our model calculations is consistent with that obtained by the optical conductivity measurements on our crystal. These agreements reveal that the magnetotransport response in SrAs₃ is determined by the single torus-shaped FS, consistent with APRES results (Fig. 1).

**Smoke-ring-type pseudospin texture**

The smoke-ring-type pseudospin texture, imprinted on the torus-shaped FSs, is confirmed by SdH oscillations. By assigning the maxima and minima of $\Delta\rho(H)$ as integers and half-integers of the Landau index (Supplementary Note 3), respectively, we plot the Landau fan diagrams

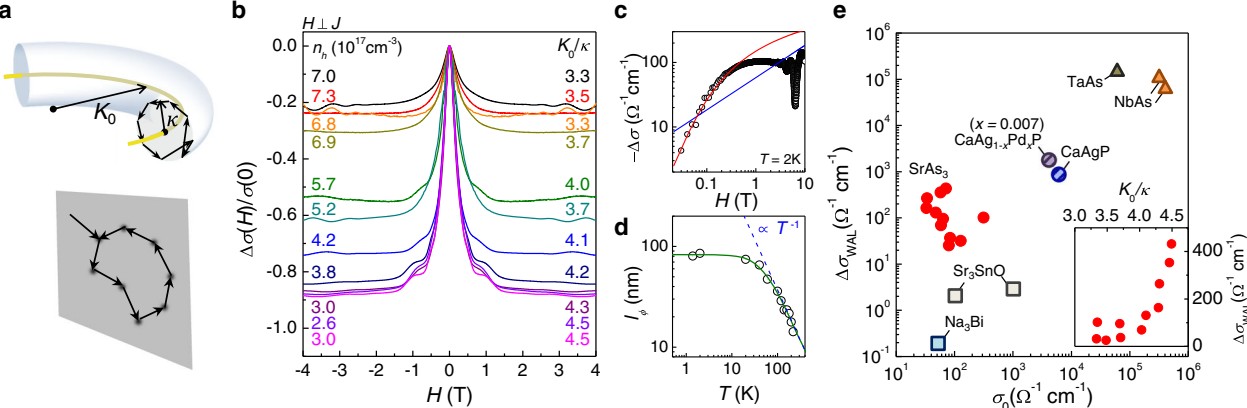

**Fig. 4 | Weak antilocalization of nodal-line fermions in SrAs$_3$. a** Back-scattering processes of nodal-line fermions on the poloidal plane of the torus-shaped Fermi surface in the momentum space (upper panel). The $\pi$ Berry flux (yellow line) along the nodal-loop leads to weak antilocalization (WAL). The corresponding diffusion of nodal-fermions in the real space is two-dimensional (lower panel), which significantly enhances the quantum interference effect. **b** The low-field transverse magnetoconductivity $\Delta\sigma(H)/\sigma(0)$, taken at $T = 2\,K$ and $H\perp J$, from eleven SrAs$_3$ crystals with different hole carrier densities ($n_h$) and the ratio ($K_0/\kappa$) between the

radii of the nodal-loop ($K_0$) and the poloidal orbit ($\kappa$). **c** The transverse magneto-conductivity $\Delta\sigma(H)$ for S1 together with the fits to the 2D WAL (red line) and 3D WAL (blue line) models. **d** Temperature-dependent phase coherence length $l_\phi$ for S1, following $T^{-1}$ dependence (blue dashed line) at high temperatures. The fit to the 2D WAL model is also shown (green solid line). **e** The excess conductivity $\Delta\sigma_{WAL}$ as a function of $\sigma_0$ for various topological semimetals. The inset shows the $\Delta\sigma_{WAL}$ of SrAs$_3$ crystals taken at 2 K with variation of the ratio $K_0/\kappa$.

for different field orientations and extract the phase offset of SdH oscillations $\phi_{SdH}$ from the interception of the linear fit (Fig. 3d–g). For both crystals (S1 and S2), $\phi_{SdH}$ as a function of the polar angle $\theta$ on the $(k_y, k_z)$ plane exhibits a clear change from −(0.3–0.4) to 0 near $\theta ∼ 10°$, when the poloidal orbit ($\alpha$) is converted to the inner toroidal ($\beta$) orbit (Fig. 3a). As the phase offset $\phi_{SdH}$ is partly determined by the Berry phase $\phi_B$, the observed change in $\phi_{SdH}$ may indicate the additional Berry phase for the poloidal orbit ($\alpha$), due to the associated the $\pi$ Berry flux and smoke-ring-type pseudospin texture. In contrast, both the inner ($\beta$) and outer ($\delta$) toroidal orbits are expected to have a zero Berry phase[41]. Consistently, for the outer toroidal orbit ($\delta$) near $H\|k_z$, we also observed the same $\phi_{SdH}$ as the inner orbit ($\beta$), as shown in Fig. 3a.

In order to clarify that the observed change in $\phi_{SdH}$ is due to the Berry phase change expected for the smoke-ring-type pseudospin texture, we consider other contributions to $\phi_{SdH}$, including the correction term for three-dimensional (3D) FS ($\phi_{3D}$) and the spin-splitting effect ($\phi_s$). The phase $\phi_0$ in Eq. (1) is determined by $\phi_B$ and $\phi_{3D}$ with a relation of $\phi_0 = -1/2 + \phi_B/2\pi + \phi_{3D}$. For hole carriers, $\phi_{3D}$ is $±1/8$ for the maximum and minimum cross-sections[42]. In addition, the spin-splitting of the Landau levels (LLs) by the Zeeman effect introduces the phase shift of SdH oscillations by $±\phi_s = ±gm^*/2m_e$, where $g$ is $g$-factor, $m^*$ is the effective mass, and $m_e$ is the free electron mass. Usually, at relatively small magnetic fields, this Zeeman spin splitting introduces the so-called spin-splitting factor $R_s = \cos(\pi gm^*/2m_e)$, and its sign change is equivalent to the phase shift of $\pi$, which often hampers precise estimation of $\phi_0$. For high-magnetic fields near the quantum limit, however, the spin splitting of LLs can be directly resolved by the additional peak splitting in SdH oscillations, which have been indeed observed in our SrAs$_3$ crystals (Fig. 3h, i). We found systematic dependence of the spin splitting of LLs on polar ($\theta$) and azimuthal angles ($\phi$), presumably due to changes in the $g$-factor and effective mass (Supplementary Note 5), as observed in other topological semimetals[29,44]. Then the extracted $\phi_s$ enables to determine the remaining phase $\phi_0$, shown in the lower panels of Fig. 3a.

In hole-doped SrAs$_3$, the poloidal orbit ($\alpha$) is expected to have an additional Berry phase ($\phi_B = \pi$) and minimum cross-section ($\phi_{3D} = -1/8$), resulting in $\phi_0 = -1/8$. On the other hand, the inner ($\beta$) and outer ($\delta$) toroidal orbits have zero Berry phase ($\phi_B = 0$) and the maximum cross-section ($\phi_{3D} = +1/8$) due to the crescent-shaped cross-section in the poloidal planes, which leads to the same $\phi_0 = -3/8$. Thus, near $\theta ∼ 10°$, when the poloidal orbit ($\alpha$) is converted to the inner

toroidal orbit ($\beta$), a phase shift by $\Delta\phi_0 = -1/4$ is expected, which is in good agreement with the observed shift $\Delta\phi_0 = -0.26(6)$ (lower panels of Fig. 3a). We note that without considering the Berry phase change, $\Delta\phi_0 = +1/4$ is expected when the $\alpha$ orbit changes to the $\beta$ orbit near $\theta ∼ 10°$, opposite to experiments.

For the azimuthal angle ($\phi$) dependence, we found that $\phi_0$ is nearly constant for different poloidal orbits around the torus-shaped FS, which is consistent with the smoke-ring-type pseudospin texture. A slight variation of $\phi_0$ with field orientation in the $(k_x, k_y)$ plane can be attributed to asymmetries in the Fermi velocity and the spin–orbit coupling (SOC), expected in SrAs$_3$ due to the low crystalline symmetry. For the Dirac node with a finite $\Delta_{SOC}$, the Berry phase is not quantized but varies from $\pi$ to zero, as described by $\phi_B = \pi\left(1 - \frac{\Delta_{SOC}}{2|\varepsilon_F|}\right)$[41]. Thus, the $\phi$-dependence in both the SOC gap ($\Delta_{SOC}$) and the $\varepsilon_F$ introduces modulation of $\phi_B$ for each poloidal cyclotron orbit. Upon rotating magnetic field from $H\|k_y$ to $H\|k_x$, the SdH frequency decreases gradually, implying that the energy position of the Dirac node corresponding to the extremal poloidal orbit becomes closer to $E_F$, reducing $|\varepsilon_F|$. This induces a slight decrease of $\phi_B$ and thus $\phi_0$, as the magnetic field approaches to $H\|k_x$. Together with the torus-shaped FS, this Berry phase evolution with magnetic field orientation provides compelling evidence for nodal-line fermions in SrAs$_3$.

## Quantum interference effect of nodal-line fermions

Now we discuss a unique quantum interference effect for the well-isolated nodal-line fermions in SrAs$_3$. Figure 4b presents the low-field magnetoconductivity, $\Delta\sigma(H)/\sigma(0)$, in transverse configuration under magnetic fields $H\|k_c$ for SrAs$_3$ crystals with different hole carrier densities ($n_h$). The sharp peak in $\Delta\sigma(H)$ is attributed to weak antilocalization (WAL) due to quantum interference of electrons with impurity scattering, as found in topological semimetals[45–48]. From the magnetoconductivity data of SrAs$_3$ (Fig. 4b) and other topological semimetals (Supplementary Fig. 11), we estimate the excess conductivity $\Delta\sigma_{WAL}$ and the semi-classical conductivity $\sigma_0$, with and without quantum interference effect, respectively, by fitting the high field data to the $H^2$ dependent conductivity from the orbital MR or the chiral anomaly effects[49,50] (Supplementary Note 6). In topological semimetals, e.g., Weyl semimetals, dominant small-angle (intravalley) scattering leads to WAL due to $\pi$ Berry phase of the back-scattering trajectories encircling a Weyl point. However, a finite large-angle (intervalley) scattering without the associated Berry phase induces the competing

weak localization (WL) and reduces the resulting $\Delta\sigma_{WAL}$[49]. Usually the large-angle scattering is more effective to reduce the semi-classical conductivity $\sigma_0$ than the small-angle scattering, the measured $\Delta\sigma_{WAL}$ is likely to decrease with lowering $\sigma_0$. Such a trend of $\Delta\sigma_{WAL}$ with variation of $\sigma_0$ is observed for various topological semimetals, as shown in Fig. 4e.

What is unique for SrAs$_3$ is the unusual magnetic field and temperature dependences of the magnetoconductivity $\Delta\sigma(H, T)$, that can be attributed to two main characters of the nodal-line fermions. First, when the poloidal orbit of radius $\kappa$ is smaller than the nodal-loop of radius $K_0$, i.e., $\kappa < K_0$,[19] the tubular FS has the local two-dimensionality in the momentum space (Fig. 4a). In SrAs$_3$, the loop radius $K_0 \sim 0.065(8)$ Å$^{-1}$ is fixed, as estimated by ARPES (Fig. 1) and SdH oscillations (Fig. 3), while reducing $n_h$ makes the tubular part of the torus-shaped FS thinner with a smaller radius $\kappa$. For SrAs$_3$ crystals with different $n_h$, we found that the SdH frequency of the poloidal orbit for $H\|c$ systematically decreases as $n_h$ reduces (Supplementary Fig. 4) and reaches the smallest size $A_F$ found among NLSM candidates so far (Supplementary Table 3). Using $F = \hbar A_F/2e\pi = \hbar\kappa^2/2e$, we estimate the averaged $\kappa$ and $K_0/\kappa \sim 3.3 - 4.5$. This is much larger than, e.g., $K_0/\kappa \sim 1.56$ of CaAgAs, a recent NLSM candidate[30], indicating the strong two-dimensional (2D) nature of the tubular FS in SrAs$_3$. Second, due to the unusual screening effect of NLSMs[16], the impurity potential is a long-range type, and the impurity scattering mainly involves with a small momentum change (small-angle scattering) at low temperatures. Therefore, the back-scattering trajectories of the electron's diffusive motion mostly lie on the 2D poloidal plane, encircling the $\pi$ Berry flux (Fig. 4a), rather than along the toroidal direction without involving the Berry flux. In this case, the dominant 2D WAL is expected to determine the magnetoconductivity $\Delta\sigma(H, T)$ of SrAs$_3$ at low-magnetic fields.

For the 2D WAL, $\Delta\sigma(H)$ is described by the Hikami-Larkin-Nagaoka model[51], roughly following the $-\ln H$ dependence, and the temperature-dependent phase coherence length $l_\phi$ follows $l_\phi \propto T^{-p/2}$ with the exponents $p = 1$ or $p = 2$ due to electron–electron or electron–phonon interactions, respectively. These behaviors are clearly distinguished from the 3D behaviors with $\Delta\sigma(H) \sim -\sqrt{H}$ and $l_\phi \propto T^{-p/2}$ with exponents $p = 3/2$ or $p = 3$ for electron–electron or electron–phonon interactions, respectively (Supplementary Note 7)[49]. The stiff drop of $\Delta\sigma(H)$ of SrAs$_3$ at low-magnetic fields is well reproduced by the fit to the 2D WAL model rather than the 3D WAL model (Fig. 4c). Consistently, the temperature-dependent $l_\phi$ follows the 2D model with the exponent of $p = 2$ at high temperatures, corresponding to the 2D electron–phonon interactions. The temperature dependence of $l_\phi$ is well reproduced by the fit to the equation, $1/l_\phi^2 = 1/l_{\phi0}^2 + A_{ep}T^2$, where $l_{\phi0} = 83(1)$ nm is the zero-temperature dephasing length and $A_{ep} = 7.0(6) \times 10^{-8}$ nm$^{-2}$K$^{-2}$ is the coefficient for electron–phonon scattering (Fig. 4d)[52]. Furthermore, since the key parameter for describing the local 2D nature of the torus-shaped FS is the ratio between the radii of the poloidal orbit ($\kappa$) and the nodal-loop ($K_0$), systematic variation of $\Delta\sigma_{WAL}$ is expected with variation of the ratio $K_0/\kappa$. As $\kappa$ becomes close to $K_0$, the contribution of weak localization by scattering along the toroidal direction without associated with Berry flux becomes sizable. Then the competition between WAL and WL determines the size of $\Delta\sigma_{WAL}$, leading to increase of $\Delta\sigma_{WAL}$ with the ratio $K_0/\kappa$, in good agreement with experiments (the inset of Fig. 4e). These results strongly indicate the 2D nature of the WAL induced by nodal-line fermions in SrAs$_3$.

## Discussion
The quantum transport signatures of nodal-line fermions, quantum oscillations and quantum interference presented in this work, consistently evince the dominant transport of nodal-line fermions in slightly hole-doped SrAs$_3$ crystals without any sizable contribution from other topologically trivial states at the Fermi level. There are several questions remained to be investigated, including quantitative understanding on the competing WAL and WL processes upon varying $K_0/\kappa$ and observation of the possible crossover between them[19]. Nevertheless, our findings highlight SrAs$_3$ as a unique platform of nodal-line fermions with the thinnest tubular FS and the largest $K_0/\kappa$ among the NLSMs candidates and thus establish SrAs$_3$ as a desirable system for studying various unique transport phenomena of nodal-line fermions, theoretically proposed but not yet realized in experiments[17–22]. Our study also emphasizes that precise control of the size difference between the radii of the nodal-loop and the poloidal cross-section is crucial for unveiling the otherwise hidden transport signature of nodal-line fermions. These findings provide a guideline for designing NLSMs suitable for novel topological electronic applications, by tuning the chemical doping or external perturbations such as strain or pressure.

## Methods
### Single-crystal growth and characterization
Single crystals of SrAs$_3$ were grown by the Bridgman method (Supplementary Note 1). The resistivity of single crystals was measured using the standard six-probe method with a Physical Property Measurement System (PPMS-14T, Quantum Design) to measure the in-plane and Hall resistivities.

### Angle-resolved photoemission spectroscopy
ARPES experiments were carried out with the Beamline 4.0.3, Advanced Light Source (Supplementary Note 2). The ARPES endstation (MERLIN) is equipped with a hemispherical electron analyzer. The energy and momentum resolutions were better than 20 meV and 0.01 Å$^{-1}$, respectively. We used the photon energy of 30–125 eV with linear-horizontal polarization. Samples were cryogenically cooled to 30–40 K and cleaved in the ultrahigh vacuum chamber with the base pressure of $1.5 \times 10^{-11}$ torr.

### Magnetotransport property measurements at high-magnetic fields
Shubnikov de Haas oscillations of SrAs$_3$ were measured using the magnetoresistivity measurements in high-magnetic fields up to 31.6 T in National High Magnetic Field Laboratory (NHMFL), Tallahassee and up to 56.7 T in International MegaGauss Science Laboratory at the Institute for Solid State Physics (ISSP), University of Tokyo (Supplementary Note 3).

### Electronic structure calculations
Electronic structures were calculated using WIEN2K code[53], which uses a full-potential augmented plane base method. The Perdew–Burke–Ernzerhof (PBE) generalized gradient approximation (GGA) was used for the exchange-correlation functional[54] and spin–orbit coupling (SOC) was included in the calculations. The modified Becke–Johnson potential (mbJ) was also employed to overcome the shortcoming of the PBE-GGA method in the underestimation of the band gap[55] (Supplementary Note 2). Two-thousand k-points were used for self-consistent calculations.

## Data availability
The data that support the findings of this study are available from the corresponding authors on request.

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

## Acknowledgements

The authors would like to thank H. W. Lee, H. W. Yeom, E. J. Choi, and M. Kang for fruitful discussions. We would also thank H. G. Kim at the Pohang Accelerator Laboratory (PAL) for the technical support. This work was supported by the Basic Science Research Program (No. 2021R1I1A1A01060209 and No. NRF-2022R1A2C3009731), BrainLink program (No. 2022H1D3A3A01077468), the Max Planck POSTECH/Korea Research Initiative (Grant No. 2022M3H4A1A04074153 and 2020M3H4A2084417), funded by the Ministry of Science and ICT through the National Research Foundation (NRF) of Korea. This work is also supported by the Institute for Basic Science (IBS) through the Center for Artificial Low Dimensional Electronic Systems (no. IBS-R014-D1) and by Samsung Advanced Institute of Technology (SAIT). S.C. and K.S.K. were supported by the NRF (Grants No. NRF-2021R1A3B1077156, NRF-2017R1A5A1014862, NRF-2020K1A3A7A09080364, NRF-RS-2022-00143178). W.K. acknowledges the support from NRF (Grants No. 2018R1D1A1B07050087 and 2018R1A6A1A03025340). A portion of this work was performed at the National High Magnetic Field Laboratory, which is supported by the National Science Foundation Cooperative Agreement No. DMR-1644779 and the State of Florida.

## Author contributions

H.K and J.S.K. conceived the experiments. H.K. synthesized the bulk single crystals. H.K., J.M.O., C.I.K, Y.K., K.K., Y.J.J., W.K., and E.S.C. carried out the transport property measurements under high-magnetic fields. S.C., K.S.K. performed angle-resolved photoemission spectroscopy and the analysis. B.G.J. and J.H.S. carried out electronic structure calculations. W.J.C. performed the structural investigation. H.K., K.S.K., and J.S.K. co-wrote the manuscript. All authors discussed the results and commented on the paper.

## Competing interests

The authors declare no competing interests.
