## [Peer Review File · Nature Communications]

REVIEWER COMMENTS

Reviewer #1 (Remarks to the Author):

The authors study magneto-transport on a nodal-line semimetal SrAs₃. By using ARPES, first, the authors identify that the nodal-line states are isolated from other bands. Then, they performed detailed magneto-transport including SdH and weak antilocalization effects. The key result is the SdH effect being consistent with the topology of isolated torus-shaped band and a large antilocalization effect which authors attributes also to the unique geometry of the band structure. The quality of experimental data is high, and the results are explained well. Figures are nicely displayed and help quick understanding of the key results.

In my opinion the paper delivers highly interesting results. The physics of low-carrier density nodal line semimetals are not easily captured only by a spectroscopic technique like ARPES. Here, by using transport which is more sensitive to low-energy physics, authors show that a large antilocalization could result from unique geometry of torus – essentially, the closed-loop around a poloidal plane giving rise to an efficient pick-up of a π phase required for antilocalization. This adds yet another example that a quantum interference effect can be a powerful probe for a geometrical phase underlying quantum materials.

I have following comment and question on the current manuscript.

1. Although the authors use $\Delta\rho(H)/\rho(0)$ to quantify a localization effect, this is somewhat misleading. The size of (anti)localization should be measured in the unit of change in conductance, $\Delta\sigma$ (e^2/h). Since the quantum interference always has a magnitude determined by quantum conductance multiplied by some degeneracy factor, if nothing else changes, high resistivity samples show “large effect”. This is not that antilocalization effect became large, but only is an artifact of plotting a phenomenon whose magnitude is pre-set by quantum conductance in terms of relative resistance change. Therefore, I think Figure 4b and 4c require revision.

2. Is other causes of positive magnetoresistance beyond antilocalization examined and rule out? The most prevalent could be classical magnetoresistance from orbital effects (naively, proportional to B^2). The lack of large negative magnetoresistance (chiral anomaly) is also intriguing. It would be helpful if authors add a brief remark on how they reject other causes of positive MR and potentially on chiral anomaly. This also relates to Fig. 4C. I presume positive MR from these materials are assigned to WAL by current authors, but how well are other origins ruled out? If original papers

assign positive MR to WAL, then I am more comfortable seeing in this plot, but if that is not the case, more careful assessment may be needed.

Reviewer #2 (Remarks to the Author):

The authors have performed systematical studies on SrAs₃, a compound proposed to be a nodal line semimetal having line-nodes close enough to Fermi energy. They have carried out the ARPES measurement for its band structure, SdH oscillation for the Fermi surface, and magnetoresistivity for the quantum interference effect due to π Berry phase. The band structure and the topology of Fermi surface have been well elaborated with first-principles calculation. These consistent and systematical efforts have presented strong evidences that SrAs₃ is indeed a nodal-line semimetal as proposed in Ref. 33. I think this work will be highly welcome and stimulate more subsequent works. I'd like to suggest its publication and I have only two minor comments:

1, Supplementary Note 1:

The mirror plane is NOT critical to have a nodal-ring state in SrAs₃. The mirror plane constrains the line nodes to be in the plane. If mirror symmetry is broken, as shown in Ref. 33, the line nodes still exist.

2, about the positive magneto-resistivity(MR)

It is argued that within external magnetic field, the nodal line will decay to Weyl nodes and chiral anomaly effect will lead to negative MR, which competes against WAL effect (i.e., positive MR). Such competition is observed in TaAs (see PRX 5, 031023 (2015)): positive MR at low magnetic field and negative MR at moderate one. I'd like to suggest the authors think about this.

Reviewer #3 (Remarks to the Author):

The manuscript reports on an electrical transport investigation of the nodal-line semimetal candidate SrAs₃. The magnetoresistance is extensively studied for multiple orientations of the magnetic field. Quantum oscillations are analysed and compared to DFT calculations and a simple linearised band model near the Y point. The data presented are of high quality and the conclusion of the material having a double torus-like Fermi surface is reasonably well supported by the experiment.

However, two major points of the paper are not grounded. Therefore, I would not recommend publication of the article in its present state.

The two major points of critique

The phase of quantum oscillations isn't only determined by the phase of the cosine in equation (1). The spin-splitting factor R_S in this formula can become negative, depending on the g-factor, and this is equivalent to a phase shift of π . Therefore, as long as the g-factor isn't known (and in this compound, it isn't), no conclusion about the Berry phase can be drawn from the observed phase of the quantum oscillations. This is an error that is very broadly made in the literature, claiming evidence for topological properties, but in most cases it isn't as simple as it seems. Therefore, figure 3d and e should be removed or rather interpreted as a simple experimental fact that cannot tell us anything about the topology.

The paper claims that the large positive magnetoresistance up to 2T is explained by weak anti localisation. What are the arguments that the classical orbital effect is not responsible for the experimental signature? Is it the dominance by 2 orders of magnitude of the hole carrier density? What is the current direction with respect to field? The MR should be studied in the longitudinal configuration, in order to rule out the orbital effect.

More comments:

— Why are the quantum oscillations (the angle dependence of the frequencies) not compared to the DFT calculations instead of the model Hamiltonian? The k-mesh for the DFT seems to me too coarse for such a small Fermi surface. Other semimetals require $300 * 300 * 300$ points in the Brillouin zone and even then the results have to be interpolated to get nice Fermi surfaces.

The angular dependence of QO frequencies should be strongly dependent on the hole carrier density (doping) of the respective samples. Is this confirmed by the experiment? Can you determine the Fermi energy using these data for different samples?

— What was the current direction with respect to the magnetic field? This is important when discussing magnetoresistance features.

— Some sentences were not clear and should be revised:

Page 5 first abstract : ...”which we used for the magneto transport measurements as discussed below.” Did you use the samples with smaller electron density? Then this should be rephrased. New sentence: “ These samples were used ...”

Second paragraph: “For a torus shaped FS... $H \parallel k_x - k_y$ ”. The field direction is a bit misleading, looks like k_x minus k_y .

— In Fig. 2d-f, it would be helpful to have the field orientation in the figures. In Fig. 2e, the inset should also show, which of the two oscillation amplitudes is shown.

Response to Reviewer #1's comments

Q1-1. *Although the authors use $\Delta\rho(H)/\rho(0)$ to quantify a localization effect, this is somewhat misleading. The size of (anti)localization should be measured in the unit of change in conductance, $\Delta\sigma$ (e^2/h). Since the quantum interference always has a magnitude determined by quantum conductance multiplied by some degeneracy factor, if nothing else changes, high resistivity samples show “large effect”. This is not that antilocalization effect became large, but only is an artifact of plotting a phenomenon whose magnitude is pre-set by quantum conductance in terms of relative resistance change. Therefore, I think Figure 4b and 4c require revision.*

A1-1. We really appreciate the reviewer's constructive comments. As the reviewer pointed out, the size of weak antilocalization (WAL) in three-dimensional (3D) metals is determined by quantum conductance e^2/h , a degeneracy factor (N), the elastic mean free path (l_e), and the phase coherence length (l_ϕ) as described by $\Delta\sigma_{\text{WAL}}=(Ne^2/h\pi^2)(1/l_e-1/l_\phi)$ [Rev. Mod. Phys. **57**, 287 (1985)]. Because the phase coherence length (l_ϕ) is larger than the mean free path (l_e) at low temperatures, the size of WAL is inversely proportional to l_e and thus become larger in highly resistive samples, as the reviewer mentioned. However, in topological semimetals, e.g. Weyl semimetals, the size of WAL is very sensitive to the relative strength of large-angle (intervalley) scattering [Phys. Rev. B **92**, 035203 (2015)]. This is because the small-angle (intravalley) scattering leads to the WAL due to π Berry phase of the backscattering trajectories encircling a Weyl point, while the large-angle (intervalley) scattering without the associated Berry phase induces the competing weak localization (WL). In most topological semimetals, the WAL by small-angle scattering usually dominates over the WL by large-angle scattering, and the resulting magnetoconductivity varies significantly with detailed balance between small-angle and large-angle scattering processes. Roughly speaking, the excess conductivity $\Delta\sigma_{\text{WAL}}$ due to the WAL is suppressed with the large-angle scattering. Because the large-angle scattering is also effective to reduce the semiclassical conductivity σ_0 , the measured $\Delta\sigma_{\text{WAL}}$ is expected to decrease in highly resistive samples with a small σ_0 . This is opposite to the simple argument for conventional 3D disordered metals, but well captures the general trend of $\Delta\sigma_{\text{WAL}}$ as a function of the semi-classical conductivity σ_0 in topological semimetals, as shown in the revised Supplementary Fig. S9q.

In nodal line semimetals like SrAs₃, two competing quantum interference processes, related to WAL and WL, contribute to electronic conduction, depending on the type of the interference paths, either encircling the nodal loop or not [Phys. Rev. Lett. **122**, 196603 (2019)]. The balance of these opposite contributions determines the excess conductivity $\Delta\sigma_{\text{WAL}}$, similar to Weyl or Dirac semimetals. When the radius of the poloidal orbit κ of the torus-shaped Fermi surface (FS) is much smaller than the radius of the nodal loop K_0 ($\kappa \ll K_0$), the backward scattering trajectories associated with small-angle scatterings mostly lie on the poloidal plane, encircling the π Berry flux. Thus the local two-dimensionality of the tubular FS leads to dominant WAL with a large $\Delta\sigma_{\text{WAL}}$. However, as κ becomes close to K_0 , the scattering along the toroidal direction becomes significant and introduces the WL contribution. Due to these competing localization processes, the size of the observed $\Delta\sigma_{\text{WAL}}$ is expected to increase with the ratio K_0/κ , which is in good agreement with the doping-dependent magnetoconductivity, as shown in Fig. 4b and the inset of Fig. 4c.

FIG. S9. Weak antilocalization analysis on various topological semimetals. **a**, The field dependent conductivity ratio $\Delta\sigma(H)/\sigma(0)$ in SrAs₃. **b–p**, The field dependent conductivity $\sigma(H)$ in various topological semimetals at low-temperature. Back-ground fitting results are presented with dashed line (**a**) or red line (**b–p**). **q**, The excess conductivity $\Delta\sigma_{\text{WAL}}$ as a function of σ_0 for various topological semimetals. The linear trend between $\Delta\sigma_{\text{WAL}}$ and σ_0 is indicated by the shaded line.

Based on these observations, we found that although the small-angle scattering associated with π Berry flux is the main origin of the WAL in topological semimetals, the relative strength of the large-angle scattering is important to determine the size of $\Delta\sigma_{\text{WAL}}$ as well as σ_0 . Moreover,

FIG. 4. Large weak antilocalization of nodal-line fermions in SrAs₃. **a**, Backscattering processes of nodal-line fermions on the poloidal plane of the torus-shaped Fermi surface in the momentum space (upper panel). The π Berry flux (yellow line) along the nodal loop leads to weak antilocalization (WAL). The corresponding diffusion of nodal-fermions in the real space is two-dimensional (lower panel), which significantly enhances the quantum interference effect. **b**, The low-field magnetoconductivity ratio $\Delta\sigma(H)/\sigma(0)$, taken at 2 K, from seven SrAs₃ crystals with different hole carrier densities (n_h) and the ratio (K_0/κ) between the radii of the nodal loop (K_0) and the poloidal orbit (κ). **c**, The ratio between the excess WAL contribution $\Delta\sigma_{\text{WAL}}$ and semiclassical conductivity σ_0 ($\Delta\sigma_{\text{WAL}}/\sigma_0$) for various topological semimetals with different carrier densities (n), including Dirac (square), Wyle (triangle), and nodal line (circle or ellipse) semimetals. The inset shows the $\Delta\sigma_{\text{WAL}}/\sigma_0$ of SrAs₃ crystals taken at 2 K with variation of the ratio K_0/κ .

when topologically-trivial bands coexist at the Fermi level, they significantly contribute to the semiclassical conductivity σ_0 but not to the excess conductivity $\Delta\sigma_{\text{WAL}}$. These observations suggest that the conductivity ratio $\Delta\sigma_{\text{WAL}}/\sigma_0$ provide a measure of the relative importance of the π Berry flux in quantum interference in electronic transport of topological semimetals.

In the revised manuscript, we compared $\Delta\sigma_{\text{WAL}}/\sigma_0$ of various topological semimetals including our SrAs₃ samples (Fig. 4c). From the magnetoconductivity $\sigma(H)$ data of SrAs₃ (Fig. 4b) and other topological semimetals (Supplementary Fig. S9), we observed a sharp peak at zero magnetic field, a hallmark of the dominant WAL effect. By fitting the high field data to the H^2 dependent conductivity of $\sigma(H)$ due to the orbital magnetoresistance or the chiral anomaly effects, we estimate both $\Delta\sigma_{\text{WAL}}$ and σ_0 , as described in the revised Supplementary Note 6. As shown in the revised Fig. 4c, we found that slightly hole-doped SrAs₃ has the largest $\Delta\sigma_{\text{WAL}}/\sigma_0$. These observations confirm that nodal-line fermions are responsible for electronic transport of SrAs₃, consistent with the ARPES (Fig. 1e) and quantum oscillations (Fig. 3), and exhibit large quantum interference contribution to the electronic conduction, as theoretically predicted in Ref. 19.

In the revised manuscript, we added a paragraph providing detailed discussion on WAL of topological semimetals, as mentioned above. The magnetoconductivity data $\sigma(H)$ and the fit to the background H^2 conduction model, used to extract the $\Delta\sigma_{\text{WAL}}$ and σ_0 for various topological semimetals, are presented in Fig. 4b and Supplementary Fig. S9.

Q1-2. *Is other causes of positive magnetoresistance beyond antilocalization examined and rule out? The most prevalent could be classical magnetoresistance from orbital effects (naively, proportional to B^2). The lack of large negative magnetoresistance (chiral anomaly) is also intriguing. It would be helpful if authors add a brief remark on how they reject other causes of positive MR and potentially on chiral anomaly. This also relates to Fig. 4C. I presume positive MR from these materials are assigned to WAL by current authors, but how well are other origins ruled out? If original papers assign positive MR to WAL, then I am more comfortable seeing in this plot, but if that is not the case, more careful assessment may be needed.*

A1-2. We do appreciate the reviewer's helpful comments. As the reviewer pointed out, not only the weak antilocalization (WAL) but also the orbital magnetoresistance (MR) and chiral anomaly effect can contribute to magnetoresistance. It has been well established that in the transverse configuration $H \perp J$, the orbital MR is significant in the absence of the chiral anomaly effect, while the chiral anomaly effect becomes dominant with much suppressed orbital MR in the longitudinal configuration $H \parallel J$. Therefore, in both configurations, we need to extract the WAL contribution from the others' contribution. As the reviewer pointed out, both the orbital MR and the chiral anomaly effects follow H^2 dependence with opposite sign, which is often negligible at low magnetic fields. This contrasts to $\sim \sqrt{H}$ or $\sim \ln H$ dependence of the WAL, dominant at low magnetic fields. As presented in Supplementary Fig. S9 for various topological semimetals, the WAL at low magnetic fields produces a sharp peak in $\Delta\sigma(H)/\sigma(0)$ and can be easily distinguished from the background H^2 dependent conductivity $\sigma(H)$ from the orbital MR or the chiral anomaly effects. Taking into account the two-channel conduction model $\sigma(H) = \Delta\sigma_{\text{WAL}}(H) + \sigma_0(H)$, we estimate the background H^2 dependent conductivity $\sigma_0(H)$ by fitting the high field data to $\sigma_0(H) - \sigma_0 \sim H^2$, as described in Supplementary Note 6. Then we extracted the WAL contribution, $\Delta\sigma_{\text{WAL}}$ and the semi-classical conductivity σ_0 at zero magnetic field (Supplementary Fig. S9), which allows us to compare $\Delta\sigma_{\text{WAL}}/\sigma_0$ of various topological semimetals (Fig. 4c).

In our experiments on SrAs₃, we used the transverse configuration $H \perp J$, rather than the longitudinal configuration $H \parallel J$ to obtain the magnetoconductivity data $\sigma(H)$. Thus the

FIG. R1. **a**, The Brillouin zone of SrAs₃ and orthogonal basis vectors k_x , k_y and k_z . The nodal ring (red circles) is located on the (k_y, k_z) plane, centered at the Y point of Brillouin zone. **b-d**, The schematic torus-shaped Fermi surfaces with different directions of the magnetic field H and current J in the longitudinal (**b**, **c**) or transverse (**d**) configurations. The toroidal (**b**) or poloidal (**c,d**) planes, normal to the external magnetic field, are indicated by red circles, on which quantum interference is strongly suppressed by magnetic field-induced dephasing. The poloidal planes, mostly contributing to electronic conduction, are indicated by blue circles in **c,d**. In **b**, all poloidal planes around the torus-shaped Fermi surface equally contribute to electronic conduction.

contribution of the chiral anomaly is absent in our experiments. For nodal-line semimetals, the transverse magnetoconductivity ($H \perp J$) is more suitable to probe the WAL effect than the longitudinal conductivity ($H \parallel J$). In SrAs₃, the nodal-loop plane is perpendicular to the ab -plane (Fig. 1d or Figs. 2a and b), and the in-plane current direction can be aligned to the k_z or k_x axis of the torus-shaped Fermi surface (FS), as illustrated in Fig. R1. Then for the longitudinal magnetoconductivity ($H \parallel J$), there are two possible configurations, $H \parallel J \parallel k_z$ (Fig. R1b) and $H \parallel J \parallel k_x$ (Fig. R1b). We note that the backscattering trajectories of diffusive electrons, responsible for the WAL, lie on the two-dimensional poloidal planes of the torus-shaped FS, encircling the π Berry flux. In order to estimate the WAL contribution $\Delta\sigma_{\text{WAL}}$ from the magnetoconductivity measurements, external magnetic fields should induce dephasing of diffusive electrons in the poloidal planes and thus need to be applied perpendicular to the poloidal plane. Then magnetic field along the k_z direction ($H \parallel J \parallel k_z$), parallel to poloidal planes, is not suitable to probe the WAL effect of diffusive electrons (Fig. R1b).

For magnetic fields along the k_x axis ($H \parallel J \parallel k_x$), dephasing of diffusive electrons occurs in the poloidal planes, normal to the k_x axis (red circles in Fig. R1c). However, diffusive electrons in these poloidal planes contribute much less to electronic conduction ($J \parallel k_x$), due to small dispersion of the tubular FS along the k_x axis, than the other poloidal planes normal to the k_y axis (blue circles in Fig. R1c). Therefore, the longitudinal magnetoconductivity measurements cannot properly capture the WAL effect. On the contrary, in the transverse configuration ($H \perp J$), for example, with $H \parallel k_y$ and $J \parallel k_x$ (Fig. R1d), diffusive electrons in the poloidal planes in the (k_x, k_z) plane contribute mostly to conduction and is also most significantly affected by magnetic-field-induced dephasing. This means that the WAL effect of the nodal-line semimetals can be well probed by the transverse magnetoconductivity measurements. Recently, in a nodal-line semimetal candidate CaAg(As,P), the larger WAL effect has been observed in the transverse configuration than in the longitudinal one [Phys. Rev. B **102**, 115101 (2020)]. Therefore, in this work, we employed the transverse configuration $H \perp J$ with $H \parallel k_x$, to measure the WAL of SrAs₃.

In the revised manuscript, we provided the detailed analysis to quantify the WAL using the magnetoconductivity data for various topological semimetals, plotted in Fig. 4c. Also the configuration of the magnetic field and current is clearly stated in the main text and in the caption of the revised Fig. 4.

Response to Reviewer #2's comments

Q2-1. *The mirror plane is NOT critical to have a nodal-ring state in SrAs₃. The mirror plane constrains the line nodes to be in the plane. If mirror symmetry is broken, as shown in Ref. 33, the line nodes still exist.*

A2-1. We highly appreciate the reviewer's helpful comments. We agree with the reviewer that the existence of the mirror plane is not critical to have a nodal-ring in SrAs₃. In the revised manuscript, we clearly state that the nodal line in SrAs₃ is accidental and not related to the mirror symmetry.

Q2-2. *About the positive magneto-resistivity(MR), it is argued that within external magnetic field, the nodal line will decay to Weyl nodes and chiral anomaly effect will lead to negative MR, which competes against WAL effect (i.e., positive MR). Such competition is observed in TaAs (see PRX 5, 031023 (2015)): positive MR at low magnetic field and negative MR at moderate one. I'd like to suggest the authors think about this.*

A2-2. We appreciate the reviewer's helpful suggestion. As the reviewer pointed out, it has been well established that the chiral anomaly effect leads to the negative longitudinal magnetoresistance (MR) in Weyl or Dirac semimetals, which competes with the WAL in the longitudinal configuration $H \parallel J$. In our experiments on SrAs₃, we used the transverse configuration $H \perp J$, rather than the longitudinal configuration $H \parallel J$. Thus the contribution of the chiral anomaly is absent in our WAL experiments (Fig. 3b).

For nodal-line semimetals, the transverse magnetoconductivity ($H \perp J$) is more suitable to probe the WAL effect than the longitudinal conductivity ($H \parallel J$). In SrAs₃, the nodal-loop plane is perpendicular to the ab -plane (Fig. 1d or Figs. 2a and b), and the in-plane current direction can be aligned to the k_z or k_x axis of the torus-shaped Fermi surface (FS), as illustrated in Fig. R1. Then for the longitudinal magnetoconductivity ($H \parallel J$), there are two possible configurations, $H \parallel J \parallel k_z$ (Fig. R1b) and $H \parallel J \parallel k_x$ (Fig. R1b). We note that the backscattering trajectories of diffusive electrons, responsible for the WAL, lie on the two-dimensional poloidal planes of the torus-shaped FS, encircling the π Berry flux. In order to estimate the WAL contribution $\Delta\sigma_{\text{WAL}}$ from the magnetoconductivity measurements, external magnetic fields should induce dephasing of diffusive electrons in the poloidal planes and thus need to be applied perpendicular to the poloidal plane. Magnetic field along the k_z direction ($H \parallel J \parallel k_z$), parallel to poloidal planes, is not suitable to probe the WAL effect of diffusive electrons (Fig. R1b).

For magnetic fields along the k_x axis ($H \parallel J \parallel k_x$), dephasing of diffusive electrons occurs in the poloidal planes, normal to the k_x axis (red circles in Fig. R1c). However, diffusive electrons in these poloidal planes contribute much less to conduction ($J \parallel k_x$), due to small dispersion of the tubular FS along the k_x axis, than the other poloidal planes, normal to the k_y axis (blue circles in Fig. R1c). Therefore, the longitudinal magnetoconductivity measurements cannot properly capture the WAL effect. On the contrary, in the transverse configuration ($H \perp J$), for example, with $H \parallel k_y$ and $J \parallel k_x$ (Fig. R1d), diffusive electrons in the poloidal planes in the (k_x, k_z) plane contribute mostly to conduction and is also most significantly affected by magnetic-field-

FIG. R1. **a**, The Brillouin zone of SrAs₃ and orthogonal basis vectors k_x , k_y and k_z . The nodal ring (red circles) is located on the (k_y, k_z) plane, centered at the Y point of Brillouin zone. **b-d**, The schematic torus-shaped Fermi surfaces with different directions of the magnetic field H and current J in the longitudinal (**b, c**) or transverse (**d**) configurations. The toroidal (**b**) or poloidal (**c,d**) planes, normal to the external magnetic field, are indicated by red circles, on which quantum interference is strongly suppressed by magnetic field-induced dephasing. The poloidal planes, mostly contributing to electronic conduction, are indicated by blue circles in **c,d**. In **b**, all poloidal planes around the torus-shaped Fermi surface equally contribute to electronic conduction.

induced dephasing. This means that the WAL effect of the nodal-line semimetals can be well probed by the transverse magnetoconductivity measurements. Recently, in a nodal-line semimetal candidate CaAg(As,P), the larger WAL effect has been observed in the transverse configuration than in the longitudinal one [Phys. Rev. B **102**, 115101 (2020)]. Therefore, in this work, we employed the transverse configuration $H \perp J$ with $H \parallel k_x$, to measure the WAL of SrAs₃.

Although we mainly focused on the WAL effect in this work, the possible chiral anomaly effect in the nodal-line semimetal is highly interesting. In fact, we found a signature of negative longitudinal magnetoresistance in the case of $H \parallel J \parallel k_x$ (Fig. 2b). However, as discussed above, the chiral anomaly effect may differ significantly, depending on the orientation of $J \parallel H$ with respect to the axis of the torus-shaped FS. We think, therefore, this issue is beyond the scope of the current work, which needs to be addressed in the future experimental or theoretical studies.

Q3-1. The phase of quantum oscillations isn't only determined by the phase of the cosine in equation (1). The spin-splitting factor R_S in this formula can become negative, depending on the g -factor, and this is equivalent to a phase shift of π . Therefore, as long as the g -factor isn't known (and in this compound, it isn't), no conclusion about the Berry phase can be drawn from the observed phase of the quantum oscillations. This is an error that is very broadly made in the literature, claiming evidence for topological properties, but in most cases it isn't as simple as it seems. Therefore, figure 3d and e should be removed or rather interpreted as a simple experimental fact that cannot tell us anything about the topology.

A3-1. We really appreciate the reviewer's helpful comments. As the reviewer pointed out the phase offset of magnetic quantum oscillations ϕ_{SdH} is sensitive to the spin-splitting of the Landau levels (LLs) by the Zeeman effect, and the Berry phase contribution (ϕ_B) can only be determined after identifying the spin splitting contribution $\phi_s = gm^*/2m_e$, where g is g -factor, m^* is the effective mass, and m_e is the free electron mass, as described by Eq. (2) in the revised manuscript. Usually, at relatively small magnetic fields, the spin-splitting effect introduces the so-called spin-splitting factor $R_s = \cos(\pi gm^*/2m_e)$, and its sign change is equivalent to the phase

FIG. 3. Toroidal Fermi surface and Berry phase evolution of SrAs₃. **a**, Angle-dependent SdH frequency (F) and the phase offset of SdH oscillation (ϕ_{SdH}) for two samples S1 (black) and S2 (red). The spin-splitting phase (ϕ_s) and the characteristic phase (ϕ_0) are also shown in the lower panels. The calculated F using the model Hamiltonian is overlaid with red lines. The corresponding extremal orbits on the torus-shaped Fermi surface are also presented for selected field orientations in the inset. **b**, Torus-shaped Fermi surface of SrAs₃ with the poloidal orbit (α) and the inner (β) and outer (δ) toroidal orbits. **c**, Poloidal cross-section of the Fermi surface (α) with pseudospin textures indicated by the arrows. **d-g**, Landau fan diagram for various field orientations with different polar (θ) angles (**d,f**) and azimuthal (ϕ) angles (**e,g**) for S1. The maxima (solid circles) and minima (open circles) of $\Delta\rho(H)/\rho(0)$ are assigned as integer and half-integer of the Landau index. **h, i**, The second derivative of $\rho(H)$, $-d^2\rho/dH^2$, as a function of the normalized F/H for various magnetic field orientations with different polar (θ) (**h**) and azimuthal (ϕ) angles (**i**) for S2. The spin splitting peaks of SdH oscillations are indicated by triangle symbols. The shaded dashed lines correspond to the spin-split Landau levels, indicated by the color-coded θ integer index and the + and - symbols.

shift of π , which prevents precise estimation of the Berry phase contribution (ϕ_B). For high magnetic fields near the quantum limit, however, the spin splitting of LLs can be directly resolved by the additional peak splitting in the Shubnikov de Hass (SdH) oscillations, which has been indeed observed in our SrAs₃ crystals (Figs. 3h and 3i of the main text, Supplementary Fig. S8). We found systematic variation of the spin splitting of LLs as a function of polar (θ) and azimuthal (ϕ) angles, indicated by the shaded lines in Figs. 3h and 3i, presumably due to changes in the g factor and effective mass, as observed in other topological semimetals [Phys. Rev. Research **2**, 012055 (2020), Nature Communications **7**, 12516 (2016)]. Then we extract the spin splitting ϕ_s , as shown in the bottom panel of Fig. 3a and determine the remaining characteristic phase ϕ_0 , which contains the Berry phase information as $\phi_0 = -1/2 + \phi_B/2\pi + \phi_{3D}$.

Based on this analysis, we clearly present the angle dependent $\phi_0(\theta, \phi)$ in Fig. 3a. In the polar angle (θ) dependent ϕ_0 , we observed a clear shift by $\Delta\phi_0 = -0.26(6)$ near $\theta \sim 10^\circ$, which can only be understood by considering the different Berry phase between the α and β orbits. As the poloidal orbit (α) is converted to the inner toroidal orbit (β) under different magnetic field orientations, one can expect the change of Berry phase from $\phi_B = \pi$ to $\phi_B = 0$ and also the change of the correction term for the 3D FS from $\phi_{3D} = -1/8$ (minimum cross-section) to $\phi_{3D} = +1/8$ (maximum cross-section). These changes of ϕ_B and ϕ_{3D} would result in $\Delta\phi_0 = -1/4$, in good agreement with experiments $\Delta\phi_0 = -0.26(6)$ as shown in the bottom panel of Fig. 3. We note that without considering the Berry phase change, $\Delta\phi_0 = +1/4$ is expected, opposite to

FIG. S8. Shubnikov-de Hass oscillations of SrAs₃ near the quantum limit. **a–d**, The second derivative of $\rho(H)$, $-d^2\rho/dH^2$, as a function of the normalized F/H for various magnetic field orientations with different polar (θ) angles (**a**, **b**) and azimuthal (ϕ) angles (**c**, **d**) for S1 (**a**, **c**) and S2 (**b**, **d**). The spin splitting peaks of SdH oscillations are indicated by triangle symbols. The shaded dashed lines correspond to the spin-split Landau levels, indicated by the color-coded integer index and the + and - symbols.

experiments.

For the azimuthal angle (ϕ) dependent ϕ_0 , we found that ϕ_0 varies slightly, in stark contrast to strong variation of ϕ . Thus, the observed strong change in ϕ_{SdH} near $\phi \sim 120^\circ$ and 240° is mainly due to the strong angle dependent spin-splitting effect. The nearly constant azimuthal angle dependence of ϕ_0 is consistent with the same pseudospin texture in poloidal orbits of the torus-shaped FS (Fig. 1c). A slight ϕ dependence of the phase ϕ_0 can be attributed to the effect of spin-orbit-coupling gap Δ_{SOC} , which reduces the Berry phase as described by $\phi_{\text{B}} = \pi(1 - \Delta_{\text{SOC}}/2|\varepsilon_{\text{F}}|)$. As the magnetic field rotates from $H \parallel k_y$ to $H \parallel k_x$, the SdH frequency decreases gradually (Fig. 3a), indicating that the ε_{F} of the corresponding poloidal orbits is reduced. This would slightly reduce ϕ_{B} and thus ϕ_0 , as the magnetic field approaches to $H \parallel k_x$, which is in good agreement with experiments.

Thanks to the reviewer's comment, we performed more careful analysis on the phase offset of SdH oscillations and present the new results, together with the related discussion in the revised manuscript, as mentioned above. These additional results and discussion provide compelling evidence for the Berry phase change in rotating magnetic fields due to the smoke-ring-type pseudospin texture on the torus-shaped FS of SrAs₃.

Q3-2. *The paper claims that the large positive magnetoresistance up to 2T is explained by weak anti localisation. What are the arguments that the classical orbital effect is not responsible for the experimental signature? Is it the dominance by 2 orders of magnitude of the hole carrier density? What is the current direction with respect to field? The MR should be studied in the longitudinal configuration, in order to rule out the orbital effect.*

A3-2. We do appreciate the reviewer's helpful comments. As the reviewer pointed out, not only the weak antilocalization (WAL) but also the orbital magnetoresistance (MR) and chiral anomaly effects can affect the magnetoresistance. It has been well established that in the transverse configuration $H \perp J$, the orbital MR is significant with absence of the chiral anomaly effect, while the chiral anomaly effect becomes dominant with much suppressed orbital MR in the longitudinal configuration $H \parallel J$. Therefore, in both cases, we need to extract the WAL contribution from the others' contribution. Because these orbital MR and the chiral anomaly effects introduce H^2 -dependent magnetoconductivity with opposite sign, these contributions are often negligible at low magnetic fields, in contrast to $\sim \ln H$ dependence of the WAL, dominant at low magnetic fields. As presented in Supplementary Fig. S9 for various topological semimetals, the WAL at low magnetic fields produces a sharp peak in $\Delta\sigma(H)/\sigma(0)$ and can be easily distinguished from the background H^2 dependent conductivity $\sigma(H)$ from the orbital MR or the chiral anomaly effect. In SrAs₃, the similar sharp peak in $\Delta\sigma(H)/\sigma(0)$ is observed in all the samples, particularly S1, A1 and A2, possessing only hole-type carriers without any signature of low-density electron carriers at low temperatures (Fig. 1, Supplementary Fig. S4b, and Supplementary Table S2) at low temperatures. These observations confirm that the sharp peak in $\Delta\sigma(H)/\sigma(0)$ in SrAs₃ samples is due to WAL, rather than the orbital MR.

In our experiments on SrAs₃, we used the transverse configuration $H \perp J$, rather than the longitudinal configuration $H \parallel J$ to obtain the magnetoconductivity data $\sigma(H)$. Thus the contribution of the chiral anomaly is absent in our experiments. For nodal-line semimetals, the transverse magnetoconductivity ($H \perp J$) is more suitable to probe the WAL effect than the longitudinal conductivity ($H \parallel J$). In SrAs₃, the nodal-loop plane is perpendicular to the ab -

FIG. S9. Weak antilocalization analysis on various topological semimetals. **a**, The field dependent conductivity ratio $\Delta\sigma(H)/\sigma(0)$ in SrAs₃. **b–p**, The field dependent conductivity $\sigma(H)$ in various topological semimetals at low-temperature. Back-ground fitting results are presented with dashed line (**a**) or red line (**b–p**). **q**, The excess conductivity $\Delta\sigma_{\text{WAL}}$ as a function of σ_0 for various topological semimetals. The linear trend between $\Delta\sigma_{\text{WAL}}$ and σ_0 is indicated by the shaded line.

plane (Fig. 1d or Figs. 2a and b), and the in-plane current direction can be aligned to the k_z or k_x axis of the torus-shaped Fermi surface (FS), as illustrated in Fig. R1. Then for the longitudinal magnetoconductivity ($H \parallel J$), there are two possible configurations, $H \parallel J \parallel k_z$ (Fig. R1b) and $H \parallel J \parallel k_x$ (Fig. R1c). We note that the backscattering trajectories of diffusive electrons, responsible for the WAL, lie on the two-dimensional poloidal planes of the torus-shaped FS, encircling the π Berry flux. In order to estimate the WAL contribution $\Delta\sigma_{\text{WAL}}$ from the magnetoconductivity measurements, external magnetic fields should induce dephasing of diffusive electrons in the poloidal planes and thus need to be applied perpendicular to the poloidal plane. Magnetic field along the k_z direction ($H \parallel J \parallel k_z$) is not suitable to probe the WAL effect of diffusive electrons (Fig. R1b).

For magnetic fields along the k_x axis ($H \parallel J \parallel k_x$), dephasing of diffusive electrons occurs in the poloidal planes, normal to the k_x axis (red circles in Fig. R1c). However, diffusive electrons in these poloidal planes contribute much less to conduction ($J \parallel k_x$), due to small dispersion of the tubular FS along the k_x axis, than the other poloidal planes, normal to the k_y axis (blue circles in Fig. R1c). Therefore, the longitudinal magnetoconductivity measurements cannot properly capture the WAL effect. On the contrary, in the transverse configuration ($H \perp J$), for example, with $H \parallel k_y$ and $J \parallel k_x$ (Fig. R1d), diffusive electrons in the poloidal planes in the (k_x, k_z) plane contribute mostly to conduction and is also most significantly affected by magnetic-field-induced dephasing. This means that the WAL effect of the nodal-line semimetals can be well probed by the transverse magnetoconductivity measurements. Recently, in a nodal-line semimetal candidate CaAg(As,P), the larger WAL effect has been observed in the transverse configuration than in the longitudinal one [Phys. Rev. B **102**, 115101 (2020)]. Therefore, in this work, we employed the transverse configuration $H \perp J$ with $H \parallel k_x$, to measure the WAL of SrAs₃.

In the revised manuscript, we provided the detailed analysis to quantify the WAL using the magnetoconductivity data for various topological semimetals, plotted in Fig. 4c. Also the configuration of the magnetic field and current is clearly stated in the main text and in the caption of the revised Fig. 4.

FIG. R1. **a**, The Brillouin zone of SrAs₃ and orthogonal basis vectors k_x , k_y and k_z . The nodal ring (red circles) is located on the (k_y, k_z) plane, centered at Y point. **b-d**, The schematic torus-shaped Fermi surfaces with different directions of the magnetic field H and current J in the longitudinal (**b**, **c**) or transverse (**d**) configurations. The toroidal (**b**) or poloidal (**c,d**) planes, normal to the external magnetic field, are indicated by red circles, on which quantum interference is strongly suppressed by magnetic field-induced dephasing. The poloidal planes, mostly contributing to electronic conduction, are indicated by blue circles in **c,d**. In **b**, all poloidal planes around the torus-shaped Fermi surface equally contribute to electronic conduction.

Q3-3. *Why are the quantum oscillations (the angle dependence of the frequencies) not compared to the DFT calculations instead of the model Hamiltonian? The k-mesh for the DFT seems to me too coarse for such a small Fermi surface. Other semimetals require 300 * 300 * 300 points in the Brillouin zone and even then the results have to be interpolated to get nice Fermi surfaces.*

A3-3. As explained in the manuscript, the low-energy states of SrAs₃ near the Fermi level are highly sensitive to the functionals used for DFT calculations (Supplementary Fig. S3). Although we were able to find reasonable agreement between the overall band structures from angle-resolved photoemission spectroscopy (ARPES) and DFT calculations using the mBJ functional, the detailed shape of a tiny Fermi surface, ~0.007% of a whole Brillouin zone, cannot be well captured by DFT calculations. Instead, we employed the model Hamiltonian taking into account all the crystalline symmetries and obtained a set of parameters, reproducing full angle-dependent quantum oscillation frequencies, the cyclotron masses obtained for three different field orientations, and the radius of the nodal-ring estimated from the ARPES. While identifying the better-performing functionals and improving the accuracy of DFT calculations for such a small Fermi surface are an important issue in the research of topological semimetals, we believe that good agreement between the low-energy model Hamiltonian and our experimental results, presented in this work, provide compelling evidence for the torus-shaped Fermi surface in our SrAs₃ crystals.

Q3-4. *The angular dependence of QO frequencies should be strongly dependent on the hole carrier density (doping) of the respective samples. Is this confirmed by the experiment? Can you determine the Fermi energy using these data for different samples?*

A3-4. As the reviewer pointed out, the quantum oscillation (QO) frequency indeed depends on the hole carrier density. Upon hole doping, the QO frequency at $H \parallel k_z$ increases systematically as shown in Supplementary Fig. S4 and Supplementary Table. 1. Using this systematic change of the SdH frequency, we estimate the Fermi energy and listed in Supplementary Table S1, as the reviewer suggested.

In this work, we present the full-angle-dependent QO results for two representative samples S1 ($n_h = 6.78 \times 10^{17} \text{ cm}^{-3}$) and S2 ($n_h = 5.22 \times 10^{17} \text{ cm}^{-3}$) with similar doping levels, which are consistent with each other. These samples are found to have a Fermi energy smaller than the band overlap energy scale and larger than the spin-orbit-coupling energy, resulting in a torus-shaped Fermi surface, which are suitable for confirming the smoke-ring-type pseudospin texture of nodal-line fermions, one of the main conclusions of this work.

For the samples with relatively low carrier densities, we expect systematic variation in full-angle-dependent QO as the reviewer pointed out. For example, if hole doping is reduced and thus the Fermi level is placed inside the SOC gap of a part of nodal loop, the resulting FS would have an elongated spheroid shape, producing a much milder angle dependence of QO frequency than observed in the torus-shaped FS. Such a change in FS topology of nodal-line fermions is a very interesting topic. Its experimental confirmation, however, requires intensive single crystals growth for the low carrier density samples and full-angle-dependent QO experiments for each sample, which is beyond the scope of our study and needs to be addressed

in the future experimental studies.

Q3-5. *What was the current direction with respect to the magnetic field? This is important when discussing magnetoresistance features.*

A3-5. As explained in **A3-2**, we employed the transverse configuration with $H \parallel k_y$ and $J \parallel k_x$. In the revised manuscript, we clearly state the current and magnetic field directions.

Q3-6. *Page 5 first abstract : ...”which we used for the magnetotransport measurements as discussed below.” Did you use the samples with smaller electron density? Then this should be rephrased. New sentence: “ These samples were used ...”*

A3-6. We thank the reviewer for careful reading on our manuscript. We used the two samples, S1 and S2 with a relatively large hole carrier density for full angle-dependent quantum oscillation measurements. To estimate the WAL effect and also to find correlation between carrier density and the quantum oscillation at $H \parallel k_y$ we used seven samples in total as listed in Supplementary Table S1. We clearly stated the samples used for full-angle dependent quantum oscillation measurements in the revised manuscript.

Q3-7. *Second paragraph: “For a torus shaped FS... $H \parallel k_x - k_y$ ”. The field direction is a bit misleading, looks like k_x minus k_y .*

A3-7. In the revised manuscript, we corrected the expression “ $H \parallel k_x - k_y$ ” to “ $H \parallel (k_x, k_y)$ plane”.

Q3-8. *In Fig. 2d-f, it would be helpful to have the field orientation in the figures. In Fig. 2e, the inset should also show, which of the two oscillation amplitudes is shown.*

A3-7. As the reviewer suggested, we clearly present the magnetic field orientation in Fig. 2 of the revised manuscript.

List of the changes

1. Considering the all reviewers' concerns about magnetoconductivity contribution other than weak antilocalization, we added the additional Supplementary Figure S9 presenting the magnetoconductivity of topological semimetals, and provided detailed information to estimate the excess WAL contribution $\Delta\sigma_{\text{WAL}}$ and semiclassical conductivity σ_0 ,

In addition, we present the ratio between the excess WAL contribution $\Delta\sigma_{\text{WAL}}$ and semiclassical conductivity σ_0 , $\Delta\sigma_{\text{WAL}}/\sigma_0$, instead of magnetoresistance, $\Delta\rho_{\text{WAL}}/\rho(H)$ in the revised Fig. 4c.

In the main text:

[Abstract, Line 32] “contribution to electric transport”

[Introduction, Line 68] “contribution to electric conduction”

[Results, Line 216-217] “magnetoconductivity, $\Delta\sigma(H)/\sigma(0)$, in transverse configuration with magnetic field $H \parallel k_c$ ”

[Results, Line 219-224] “From the magnetoconductivity data of SrAs₃ ... from the orbital MR or the chiral anomaly effects (Supplementary Note 6).”

[Results, Line 224-238] “In topological semimetals, e.g. Weyl semimetals, ...a measure of the relative importance of quantum interference associated with the π Berry flux in electronic transport of topological semimetals.”

[Results, Line 239] “ $\Delta\sigma_{\text{WAL}}/\sigma_0$ ratio”

[Results, Line 240] “yields the largest $\Delta\sigma_{\text{WAL}}/\sigma_0$ value among”

[Results, Line 249-252] “This significantly enhances the contribution ...which is consistent with experimental observations (Fig. 4c).”

[Results, Line 263] “the ratio $\Delta\sigma_{\text{WAL}}/\sigma_0$ ”

[Results, Line 263-270] “In comparison with a recent NLSM candidate ZrSiS, ...without other trivial states (Supplementary Note 7).”

[Figure 4] New results on transverse magnetoconductivity ratio $\Delta\sigma(H)/\sigma(0)$ for SrAs₃ and the ratio between the excess WAL contribution and semiclassical conductivity, $\Delta\sigma_{\text{WAL}}/\sigma_0$, for topological semimetals are presented.

In the Supplementary Information:

[First two paragraphs Supplementary Note 6] “In addition to weak antilocalization (WAL), two other contributions ...in the electronic transport of topological semimetals.”

[Supplementary Table S3] New results on the low-temperature WAL contribution in the conductivity, $\Delta\sigma_{\text{WAL}}/\sigma_0$, are included.

[Supplementary Fig. S9] New results on weak antilocalization analysis with various topological semimetals are presented.

2. As the reviewer #2 suggested, we state that the nodal line band crossing in SrAs₃ is accidental.

[Results, Line 74-75] “Band crossing accidentally occurs between”

3. Considering the reviewer #3 concerns, we present new results on the characteristic phase ϕ_0 , and spin-splitting phase ϕ_s , together with the offset ϕ_{SdH} of SdH oscillations. New results on ϕ_0 , ϕ_s , and ϕ_{SdH} of SdH oscillations are presented in the revised Fig. 3 of the main text.

In the main text:

[Results, Line 129-133] “The SdH oscillations with a single frequency F ... the phase offset of SdH oscillations ϕ_{SdH} , as discussed below.”

[Results, Line 162-166] “By assigning the maxima and minima of ... interception of the linear fit (Figs. 3d–3g).”

[Results, Line 168-172] “Because the phase offset ϕ_{SdH} is ... to have a zero Berry phase.”

[Results, Line 175-191] “In order to clarify that the observed change ...shown in the lower panels of Fig. 3a.”

[Results, Line 196-200] “Thus near $\theta \sim 10^\circ$, when the poloidal orbit (α) is converted to the inner toroidal orbit (β), ...opposite to experiments.”

[Results, Line 201-213] “For the azimuthal angle (ϕ) dependence, ...provides compelling evidence for nodal-line fermions in SrAs₃.”

In the Supplementary Information:

[Second paragraph of Supplementary Note 3] “To estimate the phase offset of SdH oscillations ϕ_{SdH} using Landau fan diagram, ...in the Landau fan diagram (Fig. S6c).”

[First two paragraphs Supplementary Note 5] “In order to estimate the phase ϕ_0 ... The phase ϕ_0 is then the same as ϕ_{SdH} from experiments.”

[Supplementary Fig. S6] New results on the comparison of SdH oscillations in ρ_{xx} and σ_{xx} are presented.

[Supplementary Fig. S8] New results on the Shubnikov-de Hass oscillations of SrAs₃ near the quantum limit are presented.

4. As requested by the reviewer #3, we included Fermi energy ϵ_F of each SrAs₃ expected from model Hamiltonian and experimental SdH oscillation frequency in Supplementary Table S1. Also, the parameters for the model Hamiltonian, obtained from the new analysis, are given in the revised Supplementary Table S2.

5. As the reviewer #3 suggested, we clearly provide the information of the samples that we

used in measurements.

[Results, Line 110-113] “These samples are used ...for investigating full angle-dependent SdH oscillations.”

6. As requested by the reviewer #3, in the revised manuscript, we used the term “ (k_x, k_y) plane” for describing the plane in the reciprocal space.

7. As requested by the reviewer #3, the cyclotron orbits for Shubnikov de Hass oscillations are illustrated in the torus-shaped Fermi surface, together with information about the external magnetic field orientation in Fig. 2 of the revised manuscript.

8. Minor changes are listed below.

In the main text:

[Results, Line 241-242] “ π Berry flux contribution in quantum interference”

[Results, Line 249] “rather than along the toroidal directions without involving Berry flux.”

[Results, Line 272-274] “The quantum transport signatures ...from other topologically trivial states at the Fermi level”

[Results, Line 277-278] “Our study also emphasizes that”

[Figure 1] The y axis scale multiplied with wrong constant is revised (Fig. 1g).

[References, Line 417-418] New reference added. “Liu, Y. *et al.* Zeeman splitting and dynamical mass generation in Dirac semimetal ZrTe₅. *Nature Communications* **7**, 12516 (2016).”

[References, Line 433-434] New reference added. “Lu, H.-Z. & Shen, S.-Q. Weak antilocalization and localization in disordered and interacting Weyl semimetals. **92**, 035203 (2015).”

[References, Line 435-436] New reference added. “Huang, X. *et al.* Observation of the Chiral-anomaly-induced negative magnetoresistance in 3D Weyl semimetal TaAs. *Phys. Rev. X* **5**, 031023 (2015).”

In the Supplementary Information:

[References] New references added. “Liu, Y. *et al.* Zeeman splitting and dynamical mass generation in Dirac semimetal ZrTe₅. *Nature Communications* **7**, 12516 (2016).”

[References] New reference added. “Huang, X. *et al.* Observation of the Chiral-anomaly-induced negative magnetoresistance in 3D Weyl semimetal TaAs. *Phys. Rev. X* **5**, 031023 (2015).”

[References] New reference added. “Lu, H.-Z. & Shen, S.-Q. Weak antilocalization and localization in disordered and interacting Weyl semimetals. **92**, 035203 (2015).”

REVIEWER COMMENTS

Reviewer #1 (Remarks to the Author):

The authors' reply and revised manuscript include some critical information related to my earlier comments, which I believe could be utilized to improve the manuscript further towards publication. As I mentioned in the previous comments, the paper does provide a new insight into quantum interference effects in transport in topological materials, whose mechanism may be novel, and my overall recommendation for the publication of this manuscript remains unchanged.

To be concise, authors seem to have partially misinterpreted my previous comment, when I mentioned that "antilocalization will look larger for low resistivity samples". I meant literally that it "look" larger when divided by zero-field resistivity (or conductivity) but in reality should not be larger in magnitude when absolute conductance change is evaluated. In Fig.4c, the authors now divide with zero-field conductance, instead of zero-field resistivity in the earlier version, which is the same thing and does not fix the issue I wanted to raise. Since the magnitude of WAL is preset by a quantum conductance (order of e^2/h) it should not change much at constant (low) temperature when a phase coherence length of the system does not change appreciably. This is indeed reflected in the authors' supplementary data (Fig. S9), where, if we limit our focus on the present data by authors (red dots), they are more or less constant with respect to σ_0 evolution (a zero-field conductance). This makes the attribution of $\Delta\sigma$ to WAL from the author actually convincing at least for me, since I expect the $\Delta\sigma$ to be more or less constant if nothing else changes (i.e., if the material system is the same). I strongly encourage replacing the current Fig.4c in the main manuscript with Fig.S9 of the supplement, since, in my opinion, the former is misleading due to a reasoning I intended to deliver in the previous comments. The plot (Fig.S9) in the supplement, on the other hand, is much more informative and delivers physics in a less misleading manner.

Reviewer #2 (Remarks to the Author):

The authors have addressed my concerns well and I recommend this revision to be accepted.

Reviewer #3 (Remarks to the Author):

The article by Kim et al was improved, but in my opinion contains still some too strong claims that are not based on the experimental facts. Many questions are open and they should be addressed before publication.

1.

Based on the mBJ, which was confirmed by ARPES as the authors say, the authors should make a statement about the size of the band overlap, the spin-orbit splitting and the tilting (within uncertainty of the calculations). This is important information in order to find out, if the model hamiltonian makes sense. Especially, the authors would find out, if the experimental signal, that consists mainly of 1-2 oscillation frequencies, is actually a superposition of 2-4 frequencies, depending on the orientation. This should be the case, if the two tori are rather close in energy and in size.

2.

If the two tori are close in size, then, the measured phase of the quantum oscillations might be influenced by the relative phase of the two underlying frequencies of the observed one, and for such low frequencies, it is very difficult to disentangle the contributions of each oscillation. It might then happen, that the phase changes due to a relative change of the oscillation amplitude. Can the authors rule out such an origin of the observed phase? I guess I would agree with the analysis if the authors said that the number of oscillations should be 2-4 depending on the orientation, but experimentally, only 1-2 are observed and if it is assumed that these are the only ones, then ... can be said about the phase.

3.

How does the splitting behave as a function of field (is it linear as expected for Zeeman?) How do the authors explain the huge anisotropy of the g-factor they find from their analysis? How does the spin-splitting appear in the Landau fan diagram? It would be very helpful to have graphs where such things could be visible and where the quality of the linear fits in the Landau fan diagram are visible.

4.

In the discussion about the localisation physics, I think the text is too much written as if all that is said is a fact. It is possible that there exist a theory that predicts the explained behaviour, but in my

opinion, these statements are far from proven and therefore some more care in wording them as possible explanations would be needed here.

5.

And there are open questions about the plot of Δ_{σ_WAL} versus σ_0 for different materials. Here, measurements from longitudinal transport measurements should be removed from this plot unless current jetting effects have carefully been ruled out in the respective measurements.

There is a prediction, that when the small-angle scattering dominates the localisation physics, the regime should be 2D and then the B-dependence is supposed to be different from the 3D behaviour. This could easily be tested, when claims about a change of regime are made.

As well, WAL and WL are in theory accompanied by a certain temperature dependence. Is this observed in any of the given materials and especially, is it observed in SrAs₃? Why not, if not?

6.

I don't understand the figure S9q. Are the authors saying that the positive trend on this curve is intrinsic? How do they then explain the negative trend for their samples? Are both intrinsic? How does this fit together? They should give an explanation of the opposite behaviour.

7.

One way to get information about the ratio of large angle scattering to small angle scattering is from comparing the mean free path from resistivity (only large angle scattering contributes) with the one from quantum oscillations (all scattering is contributing). What is this ratio in SrAs₃?

8. Other things

In line 129, the sentence is not finished.

The number of significant digits is too large for both the effective masses and the scattering times, which usually have error bars of roughly 10 percent for the mass and larger for the scattering time.

Q1-1. To be concise, authors seem to have partially misinterpreted my previous comment, when I mentioned that "antilocalization will look larger for low resistivity samples". I meant literary that it "look" larger when divided by zero-field resistivity (or conductivity) but in reality should not be larger in magnitude when absolute conductance change is evaluated. In Fig.4c, the authors now divide with zero-field conductance, instead of zero-field resistivity in the earlier version, which is the same thing and does not fix the issue I wanted to raise. Since the magnitude of WAL is preset by a quantum conductance (order of e^2/h) it should not change much at constant (low) temperature when a phase coherence length of the system does not change appreciably. This is indeed reflected in the authors' supplementary data (Fig. S9), where, if we limit our focus on the present data by authors (red dots), they are more or less constant with respect to σ_0 evolution (a zero-field conductance).

This makes the attribution of $\Delta\sigma$ to WAL from the author actually convincing at least for me, since I expect the $\Delta\sigma$ to be more or less constant if nothing else changes (i.e., if the material system is the same). I strongly encourage replacing the current Fig.4c in the main manuscript with Fig.S9 of the supplement, since, in my opinion, the former is misleading due to a reasoning I intended to deliver in the previous comments. The plot (Fig.S9) in the supplement, on the other hand, is much more informative and delivers physics in a less misleading manner.

A1-1. We appreciate the reviewer's helpful comments and suggestions. As the reviewer suggested, we revised Fig. 4e using Fig. S9 presented in the previous Supplementary information, which shows the trend of $\Delta\sigma_{\text{WAL}}$ as a function of the conductivity σ_0 for various topological semimetals. In this revised Fig. 4e, we plotted the data of eleven SrAs₃ crystals, including the results from the four additional samples (A1, A7-9). For the data from other topological semimetals, we plotted only the cases when the current jetting effect is carefully considered, following the third reviewer's suggestion (A3-6). In this revised plot, the $\Delta\sigma_{\text{WAL}}$ data of SrAs₃ nicely form a cluster and are somewhat smaller than those of other highly-conducting topological materials.

FIG. 4. Weak antilocalization of nodal-line fermions in SrAs₃. **a**, Backscattering processes of nodal-line fermions on the poloidal plane of the torus-shaped Fermi surface in the momentum space (upper panel). The π Berry flux (yellow line) along the nodal loop leads to weak antilocalization (WAL). The corresponding diffusion of nodal-fermions in the real space is two-dimensional (lower panel), which significantly enhances the quantum interference effect. **b**, The low-field magnetoconductivity ratio $\Delta\sigma(H)/\sigma(0)$, taken at 2 K, from eleven SrAs₃ crystals with different hole carrier densities (n_h) and the ratio (K_0/κ) between the radii of the nodal loop (K_0) and the poloidal orbit (κ). **c**, The transverse magnetoconductivity $\Delta\sigma(H)$ for S1 together with the fits to the 2D WAL (red line) and 3D WAL (blue line) models. **d**, Temperature-dependent phase coherence length l_ϕ for S1, following T^{-1} dependence (blue dashed line) at high temperatures. The fit to the 2D WAL model is also shown (green solid line). **e**, The excess conductivity $\Delta\sigma_{\text{WAL}}$ as a function of σ_0 for various topological semimetals. The inset shows the $\Delta\sigma_{\text{WAL}}$ of SrAs₃ crystals taken at 2 K with variation of the ratio K_0/κ .

The detailed analysis of the $\Delta\sigma_{\text{WAL}}$ data only for SrAs₃ crystals reveals a clear trend of $\Delta\sigma_{\text{WAL}}$. As shown in the inset of Fig. 4e, the absolute magnitude of $\Delta\sigma_{\text{WAL}}$ varies systematically with the ratio between the radii of the nodal loop K_0 and the poloidal orbit κ . Such a systematic variation of $\Delta\sigma_{\text{WAL}}$ is consistent with the theoretical study on nodal-line semimetals [Phys. Rev. Lett. **122**, 196603 (2019)]. In nodal line semimetals, two quantum interference processes along the poloidal and toroidal directions compete with each other, determining the size of $\Delta\sigma_{\text{WAL}}$. As the radius of the poloidal orbit κ of the torus-shaped Fermi surface (FS) becomes smaller than the radius of the nodal loop K_0 ($\kappa \ll K_0$), the backward scattering trajectories, associated with small-angle scatterings, tend to encircle the π Berry flux in the poloidal plane, making the WAL effect dominant. In contrast, as κ becomes close to K_0 , the scattering along the toroidal direction becomes significant and introduces the weak localization (WL) contribution. Thus, the absolute magnitude of the observed $\Delta\sigma_{\text{WAL}}$ is expected to increase with the ratio K_0/κ , in good agreement with experiments.

The dominant WAL process along the poloidal direction is further confirmed by the dimensionality of the observed WAL behaviors. As predicted in Ref. 19 [Phys. Rev. Lett. **122**, 196603 (2019)], when the small-angle scattering in the poloidal plane is dominant, the WAL should follow the two-dimensional (2D) behavior due to the tubular FS of nodal-line semimetal. Such prediction can be tested by examining the magnetic-field dependent conductivity $\Delta\sigma(H)$ and also the temperature-dependent phase coherence length l_ϕ . For the 2D WAL, the magneto-conductivity $\Delta\sigma(H)$ is described by the Hikami-Larkin-Nagaoka model, roughly following the $-\ln H$ dependence [Prog. Theor. Phys. **63**, 707 (1980)], and the phase coherence length follows $l_\phi \propto T^{-p/2}$ with the exponents $p = 1$ or $p = 2$ due to electron-electron or electron-phonon interactions, respectively. These behaviors are clearly distinguished from the 3D behaviors with $\Delta\sigma(H) \sim -\sqrt{H}$ and $l_\phi \propto T^{-p/2}$ with exponents $p = 3/2$ or $p = 3$ for electron-electron or electron-phonon interactions, respectively [Phys. Rev. B **92**, 035203 (2015)]. As shown in Fig. 4c, the stiff drop of $\Delta\sigma(H)$ of SrAs₃ at low magnetic fields is well fitted with the 2D WAL model ($\sim -\ln H$) rather than the 3D WAL model ($\sim -\sqrt{H}$). Furthermore, the temperature-dependence of $l_\phi \propto T^{-1}$ at high temperatures corresponds to the exponent $p = 2$ for the 2D electron-phonon interactions [J. Phys. Condens. Matter **14**, R501 (2002)]. These results support that the WAL in the bulk SrAs₃ is in excellent agreement with the theoretically-predicted 2D character of nodal-line semimetals.

In the revised manuscript, we removed the discussion on the relative size of $\Delta\sigma_{\text{WAL}}/\sigma_0$ of SrAs₃, as compared with those of other topological semimetals, following the reviewer's suggestion. Instead, using our additional experimental data and analysis, we emphasize that the unique 2D characters of WAL, expected for nodal line semimetals, are clearly observed in our SrAs₃ crystals.

Q3-1. Based on the mBJ, which was confirmed by ARPES as the authors say, the authors should make a statement about the size of the band overlap, the spin-orbit splitting and the tilting (within uncertainty of the calculations). This is important information in order to find out, if the model hamiltonian makes sense.

Especially, the authors would find out, if the experimental signal, that consists mainly of 1-2 oscillation frequencies, is actually a superposition of 2-4 frequencies, depending on the orientation. This should be the case, if the two tori are rather close in energy and in size

A3-1. We highly appreciate the reviewer's helpful comments. As we emphasized in our manuscript, the calculated electronic structures are sensitive to the exchange functionals used in the density-functional-theory (DFT) calculations. This allows us to compare the calculated band structures with the observed ARPES spectra only in a relatively wide energy scale and to identify that calculations using mBJ functionals agree reasonably with experiments. Although this comparison clearly shows that there are only two intersecting parabolic bands placed at the Fermi level, forming a nodal loop (Fig. S3), the fine details of band structures near the Fermi level show deviation from those obtained in calculations. Particularly for semimetals with a small Fermi energy, like SrAs₃, it is known quite challenging to precisely reproduce the fine details of Fermi surface shape using DFT-based calculations.

For example, using mBJ functionals, we estimated the size of the band overlap to be $\Delta \sim 300$ meV. In our recent optical spectroscopy measurements (Fig. R1), however, the band overlap energy is estimated to be $\Delta \sim 120$ meV. In nodal-line semimetals with two intersecting parabolic bands and a single nodal-loop, the interband optical transition produces characteristic frequency-dependent optical conductivity $\sigma(\omega)$ in the high frequency regime [Phys. Rev. B **96**, 155150 (2017)]. Along the k_z axis, the axial optical conductivity $\sigma_{zz}(\omega)$ shows a plateau up to the band overlap energy Δ , followed by a clear drop at higher energies ($\hbar\omega > \Delta$). On the other hand, along the k_x axis, the radial conductivity $\sigma_{xx}(\omega)$ also shows a plateau ($\hbar\omega < \Delta$) with a mild increase for $\hbar\omega > \Delta$. For our SrAs₃ single crystal, we clearly observed these characteristic behaviors in $\sigma_{zz}(\omega)$ and $\sigma_{xx}(\omega)$ in the high frequency regime, together with the Drude peak (at lowest energy) and sharp phonon peaks ($\sim 15, 20, 30$ meV) as shown in Fig. R1a. Therefore, the characteristic frequency $\Delta \sim 120$ meV, showing a clear kink in $\sigma_{zz}(\omega)$, corresponds to the band overlap energy, much smaller than $\Delta \sim 300$ meV from DFT calculations. Furthermore, the radius of nodal loop from calculations is $K_0 \sim 0.10 \text{ \AA}^{-1}$, which turns out to be larger than $K_0 \sim 0.057 \text{ \AA}^{-1}$ estimated from the ARPES results in Fig. 1i of the main text.

Bearing these observations in mind, we constructed the low-energy model Hamiltonian, taking into account the crystalline symmetries of SrAs₃, that explains the observed angle-dependent frequencies of quantum oscillations (Fig. 3a). The validity of our model Hamiltonian is checked by comparing three band parameters that were experimentally determined, including the cyclotron masses for three different field directions estimated from quantum oscillations, the radius of the nodal loop from the ARPES spectra (Fig.1), and the size of the band overlap determined by the optical spectroscopy (Fig. R1). As shown in the Table R1 below, all three characteristic

FIG. R1. Optical conductivity and band dispersions of SrAs₃ **a**, Optical conductivity at 5 K with polarization along the k_x direction (red line) and the k_z direction (green line). **b**, **c**, Band structures of the model Hamiltonian along the k_z direction (**b**) and the k_x direction (**c**).

parameters are in good agreement with our model Hamiltonian. These results strongly indicate that our model Hamiltonian correctly captures the low energy band structure near the Fermi level. We note that the size of the spin-orbit coupling (SOC) gap, located above the Fermi level, cannot be determined experimentally. However, even if we used the calculated SOC gap of $\sim 10 - 35$ meV, the Fermi surface is not significantly modified because the Fermi level is located ~ 50 meV below the nodal loop, which again suggests that our model Hamiltonian is suitable to describe the Fermi surface properties of the samples investigated in this work.

	Experiment			Model Hamiltonian	
Cyclotron mass m^*/m_e	Quantum oscillations ults	α orbit	$H \parallel k_x$	0.056(2), 0.033(1)	0.04381
			$H \parallel k_y$	0.076(5), 0.080(4)	0.07437
		β orbit	$H \parallel k_z$	0.023(1), 0.176(8)	0.246
			δ orbit	$H \parallel k_z$	0.079(3)
Nodal loop radius K_0	ARPES	0.057 \AA^{-1}		0.065 \AA^{-1}	
Band overlap energy Δ	Optical conductivity	~ 120 meV		139 meV	

TABLE. R1. Band parameters of SrAs₃ from the experimental results and the model Hamiltonian

Having established that no additional electronic bands rather than two intersecting parabolic bands, based on ARPES results, a single torus-shaped FS with spin degeneracy is expected within the entire Brillouin zone, which is identified experimentally in this work. We note that spin-splitting by antisymmetric spin-orbit interaction is forbidden in centrosymmetric SrAs₃, which contrasts to the cases of noncentrosymmetric semimetals [e.g. Phys. Rev. B **101**, 245104 (2020)] having two tori with similar sizes or shapes. The remaining possibility to introduce two tori of FSs is Zeeman spin-splitting under high magnetic fields. However, such an effect has already been taken into account in our analysis using Eq. (1), as shown in Figs. 3h and i. These observations strongly suggest that the Berry phase contribution ϕ_0 , presented in Fig. 3a, can be attributed to the pseudospin texture of a single torus-shaped FS with spin degeneracy in SrAs₃.

In order to clarify the issues discussed above, we clearly state the limitation of the DFT-based calculations for the fine details of the low-energy band structures of SrAs₃ and emphasize good agreement between our model Hamiltonian and experimentally-determined band parameters in the revised manuscript.

Q3-2. *If the two tori are close in size, then, the measured phase of the quantum oscillations might be influenced by the relative phase of the two underlying frequencies of the observed one, and for such low frequencies, it is very difficult to disentangle the contributions of each oscillation. It might then happen, that the phase changes due to a relative change of the oscillation amplitude. Can the authors rule out such an origin of the observed phase? I guess I would agree with the analysis if the authors said that the number of oscillations should be 2-4 depending on the orientation, but experimentally, only 1-2 are observed and if it is assumed that these are the only ones, then ... can be said about the phase.*

A3-2. As we explained in **A3-1**, our model Hamiltonian is consistent with the underlying crystalline symmetries and the experimentally-determined characteristics of Fermi surfaces (FSs), including angle-dependent frequencies, cyclotron masses, the size of band overlap, and the radius of nodal loop. As the reviewer pointed out, with the observed quantum oscillation data alone, we cannot completely rule out the possibility of additional tori that are accidentally close in size and shape. However, by combining with ARPES experiments and band structure calculations, we show that only two spin-degenerate parabolic bands are located at the Fermi level, forming a single-torus FS with a nodal loop in the Brillouin zone. The inversion symmetry of SrAs₃ also rules out the possible spin-splitting by the antisymmetric spin-orbit interaction, that usually split the torus into two tori in noncentrosymmetric semimetals [Phys. Rev. B **101**, 245104 (2020)]. Therefore, our set of experimental observations strongly indicates that the single-torus-shaped FS exists in SrAs₃.

Q3-3. How does the splitting behave as a function of field (is it linear as expected for Zeeman?) How does the spin-splitting appear in the Landau fan diagram? It would be very helpful to have graphs where such things could be visible and where the quality of the linear fits in the Landau fan diagram are visible.

A3-3. We do appreciate the reviewer's helpful comments. It has been known that for parabolic bands, quantum oscillations exhibit the constant Zeeman splitting when plotted as a function of $1/H$, because energy differences by Zeeman splitting and Landau level splitting depend linearly in external magnetic field H [PNAS **115**, 9145–9150 (2018)]. This is also the case for Dirac bands when $2\hbar ev_0^2 H \gg (g\mu_B H/2)^2$, where g is the g -factor, μ_B is Bohr magneton, \hbar is the reduced Plank constant, e is electron charge, and v_0 is the band velocity [PNAS **115**, 9145–9150 (2018)]. In our study, even for the maximum magnetic field of 31.6 T and the largest g factor ~ 19.1 , $2\hbar ev_0^2 H$ is one order larger than $(g\mu_B H/2)^2$, guaranteeing the validity of Eq. (1).

To further clarify this issue, we plot the Landau fan diagram, including the spin-split Landau levels, in the revised Supplementary Fig. S8 as the reviewer suggested. In Shubnikov-de Hass oscillations, two Zeeman-split peaks in the second derivative of the oscillating magnetoresistivity, $d^2\rho/dH^2$ as a function of $1/H$ are separate by a spacing of ϕ_s/F , where spin-splitting phase $\phi_s = gm^*/2m_c$ is determined by the g -factor (g) and the effective mass (m^*) with respect to free electron mass (m_c). In SrAs₃, high-field quantum oscillation data can be classified into three representative cases depending on the size of ϕ_s (Fig. S8). When the Zeeman splitting is smaller than the spin-degenerate Landau level spacing ($1/F$) *i.e.* $\phi_s \sim 0-0.3$, disorder-induced broadening makes two peaks in the oscillating magnetoresistivity merge into one peak, preventing experimental determination of the Zeeman splitting. For $\phi_s \sim 0.35-0.5$, however, the Zeeman splitting spacing becomes large enough to be detected as shown in Figs. S8b, S8e and S8h, and we assigned the middle point of the two Zeeman-split peaks with the integer Landau index. We note that the spin-splitting appears to be constant in the Landau fan diagram, consistent with the discussion above. The linear fitting in the Landau fan diagram shows excellent agreement with the corresponding R-square

FIG. S8. Landau fan diagram with Zeeman splitting in SrAs₃. **a-c,** The schematic illustrations of quantum oscillation peaks depending of the size of the spin splitting phase ϕ_s . The vertical bars under oscillation curves correspond to the Zeeman-split levels from the spin-degenerate Landau level. **d-i,** The quantum oscillations in the second derivative of resistivity, $d^2\rho/dH^2$, and the corresponding Landau fan diagram for S2. The magnetic field directions are on the (k_y, k_z) plane (**d, e, f**) or (k_x, k_y) plane (**g, h, i**).

value > 0.99 . This is also the case for larger Zeeman splitting with $\phi_s > 0.5$, in which the spin-split Landau levels with different orbital Landau indices, such as $3+$ and $4-$, become close and eventually produce one peak in $d^2\rho/dH^2$ curve. In this case, we assigned a deep as integer Landau index, which again follows a linear dependence in the Landau fan diagram with R-square > 0.99 .

Q3-4. *How do the authors explain the huge anisotropy of the g-factor they find from their analysis?*

A3-4. The size and anisotropy of the g-factor are determined by the spin-orbit coupling strength, a band gap size, and matrix element between two neighboring bands. Since the Zeeman coupling is not only involved by the spin momentum but also the orbital momentum of the Bloch states, it has been found that the g-factor can be much larger than free electron's $g = 2$ and shows significant anisotropy with respect to the crystallographic directions in topological insulator or semimetals, where multiple bands are crossed or placed closely in energy. As presented in Table R2 for two representative examples of topological materials, a topological semimetal ZrTe₅ and a topological insulator Bi₂Se₃, a large anisotropy of the g-factor, comparable with the case of SrAs₃ has been observed. To understand such a large g-factor anisotropy in SrAs₃, the detailed analysis on the interband matrix elements is required. This issue is certainly interesting and important, but we believe it is beyond the scope of this work.

	Theory / Experiment	Direction			Reference
		a -axis	b -axis	c -axis	
SrAs ₃	Experiment	2	2.6-3.3	18.7-19.1	[This work]
ZrTe ₅	Experiment	5.3	15.8-24.3	7.6	[PNAS 115 , 9145–9150 (2018)] [Phys. Rev. B 96 , 041101(R) (2017)] [Nat. Commun. 7 , 12516 (2016)] [Phys. Rev. Lett. 115 , 176404 (2015)]
	Theory	-0.12 (LDA) -0.04 (GGA) 0.08 (mBJ)	12.24 (LDA) 11.63 (GGA) 9.66 (mBJ)	-5.19 (LDA) -4.56 (GGA) -2.22 (mBJ)	[arXiv: 1512.05084]
Bi ₂ Se ₃	Experiment	18.96-19.48		27.3-29.9	[Phys. Rev. B 93 , 155114 (2016)]
	Theory	16.37 (LDA) 17.86(GGA) 26.18 (mBJ)		18.4 (LDA) 21.76(GGA) 41.8 (mBJ)	[arXiv: 1512.05084]

TABLE R2. The g-factors of various topological materials derived from experiments and calculations.

Q3-5. *In the discussion about the localisation physics, I think the text is too much written as if all that is said is a fact. It is possible that there exist a theory that predicts the explained behaviour, but in my opinion, these statements are far from proven and therefore some more care in wording them as possible explanations would be needed here.*

A3-5. As the reviewer suggested, we toned down our statements on weak antilocalization, particularly about the size comparison with other topological semimetals. Instead, we presented additional results on two-dimensional behaviors of weak antilocalization in SrAs₃, following the reviewer's suggestions [A3-6], which is consistent with the theoretical prediction. We also clearly state that there are many remaining issues that requires further theoretical and experimental investigations in the revised manuscript.

Q3-6. *And there are open questions about the plot of Delta_sigma_WAL versus sigma_0 for different materials. Here, measurements from longitudinal transport measurements should be removed from this plot unless current jetting effects have carefully been ruled out in the respective measurements.*

There is a prediction, that when the small-angle scattering dominates the localisation physics, the regime should be 2D and then the B-dependence is supposed to be different from the 3D behaviour. This could easily be tested, when claims about a change of regime are made. As well, WAL and WL are in theory accompanied by a certain

temperature dependence. Is this observed in any of the given materials and especially, is it observed in SrAs₃? Why not, if not?

A3-6. We really appreciate the reviewer's helpful comments. As the reviewer suggested, we removed the data from the studies of the longitudinal transport measurements on Sr₃SnO [Nat. Commun. **11**, 1161 (2020)], CaAgP [Phys. Rev. B **102**, 115101 (2020)], Pd-doped CaAgP [Phys. Rev. B **102**, 115101 (2020)], ZrTe₅ [Nat. Phys. **12**, 550–554 (2016)], Cd₃As [Nat. Mater. **14**, 280–284 (2015)] and ZrSiS [PNAS **114**, 2468–2473 (2017)], in which the current jetting effect has not been carefully considered and ruled out.

As the reviewer correctly pointed out, the WAL of the tubular FS of nodal-line semimetal should follow the two-dimensional (2D) behavior when the small-angle scatterings in the poloidal plane are dominant [Phys. Rev. Lett. **122**, 196603 (2019)]. Such prediction can be tested by examining the magnetic-field-dependent conductivity $\Delta\sigma(H)$ and also the temperature-dependent phase coherence length l_ϕ . For the 2D WAL, the magneto-conductivity $\Delta\sigma(H)$ is described by the Hikami-Larkin-Nagaoka model (HLN model), $\Delta\sigma(H) = -\alpha \frac{e^2}{2\pi^2\hbar} \left[\psi\left(\frac{1}{2} + \frac{B_\phi}{H}\right) - \ln\left(\frac{B_\phi}{H}\right) \right]$, where ψ is digamma function, e is electron charge, \hbar is Planck constant, $B_\phi = \hbar^2/(4el_\phi^2)$ is characteristic field associated with phase coherence length l_ϕ [Prog. Theor. Phys. **63**, 707 (1980)]. In this case, $\Delta\sigma(H)$ roughly follows $-\ln H$ dependence at low magnetic fields. In contrast, 3D WAL of topological semimetals produces $\Delta\sigma(H)$ described by $\sigma(H) \sim \frac{2e^2}{h} \int_0^{1/l} \frac{dx}{(2\pi)^2} \left[\psi\left(\frac{l_B^2}{l^2} + l_B^2 x^2 + \frac{1}{2}\right) - \psi\left(\frac{l_B^2}{l^2} + l_B^2 x^2 + \frac{1}{2}\right) \right]$ with l the mean free path and $l_B = \sqrt{\hbar/4eH}$ the magnetic length [Phys. Rev. B **92**, 035203 (2015)], which shows $-\sqrt{H}$ dependence at low magnetic fields. As shown in Fig. 4c of the main text and Fig. S11 in the Supplementary information, the stiff drop of $\Delta\sigma(H)$ in SrAs₃ is well fitted to the 2D WAL model rather than the 3D WAL model.

The temperature dependence of phase coherence length also supports the same conclusion. In the 2D WAL model, the phase coherence length follows $l_\phi \propto T^{-p/2}$ dependence with the exponents $p = 1$ or $p = 2$ due to electron-electron or electron-phonon interactions, respectively. This contrasts to the 3D WAL behavior described by different exponents, $p = 3/2$ or $p = 3$ for electron-electron or $p = 3$ for electron-phonon interactions. As shown in

FIG. 4. Weak antilocalization of nodal-line fermions in SrAs₃. **a**, Backscattering processes of nodal-line fermions on the poloidal plane of the torus-shaped Fermi surface in the momentum space (upper panel). The π Berry flux (yellow line) along the nodal loop leads to weak antilocalization (WAL). The corresponding diffusion of nodal-fermions in the real space is two-dimensional (lower panel), which significantly enhances the quantum interference effect. **b**, The low-field magnetoconductivity ratio $\Delta\sigma(H)/\sigma(0)$, taken at 2 K, from eleven SrAs₃ crystals with different hole carrier densities (n_h) and the ratio (K_0/κ) between the radii of the nodal loop (K_0) and the poloidal orbit (κ). **c**, The transverse magnetoconductivity $\Delta\sigma(H)$ for S1 together with the fits to the 2D WAL (red line) and 3D WAL (blue line) models. **d**, Temperature-dependent phase coherence length l_ϕ for S1, following T^{-1} dependence (blue dashed line) at high temperatures. The fit to the 2D WAL model is also shown (green solid line). **e**, The excess conductivity $\Delta\sigma_{\text{WAL}}$ as a function of σ_0 for various topological semimetals. The inset shows the $\Delta\sigma_{\text{WAL}}$ of SrAs₃ crystals taken at 2 K with variation of the ratio K_0/κ .

FIG. S11. Magnetoconductivity and weak antilocalization of SrAs₃ crystals. **a**, **b**, Magnetoconductivity $-\Delta\sigma(H)$ (**a**) and the normalized magnetoconductivity (**b**) for eleven SrAs₃ crystals with 2D WAL (red line) and 3D WAL (blue line). **c**, Temperature dependent magnetoconductivity of S1 with HLN equation fitting (yellow line). **d**, Temperature-dependent phase coherence length l_ϕ for S1, following T^{-1} dependence (blue dashed line) at high temperatures. The fit to the 2D WAL model is also shown (green solid line).

Fig. 4d of the main text, the temperature-dependence $l_\phi \propto T^{-1}$ at high temperatures corresponds to the exponent $p = 2$ for 2D electron-phonon interactions. In details, the temperature-dependence of l_ϕ in the 2D diffusive system can be described by the expression, $1/l_\phi^2 = 1/l_{\phi_0}^2 + A_{ee}T + A_{ep}T^2$ where l_{ϕ_0} is zero-temperature dephasing length, A_{ee} and A_{ep} are contribution factor from electron-electron and electron-phonon scattering, respectively [J. Phys. Condens. Matter **14**, R501 (2002)]. The best fit shown in the Fig. 4d of the main text reproduce nicely the experimental data, yielding $l_{\phi_0} = 83(1)$ nm, $A_{ee} \approx 0$ and $A_{ep} = 7.0(6) \times 10^{-8}$ nm² K⁻², which shows clear 2D WAL with dominant electron-phonon interaction. These results strongly suggest that the observed magnetoconductivity of the bulk SrAs₃ crystals agrees well with the theoretically-predicted 2D WAL behaviors of nodal-line semimetals.

Q3-7. *I don't understand the figure S9q. Are the authors saying that the positive trend on this curve is intrinsic? How do they then explain the negative trend for their samples? Are both intrinsic? How does this fit together? They should give an explanation of the opposite behaviour.*

A3-7. We appreciate the reviewer's helpful comments. In the revised Fig. 4e (Fig. S9 presented in the previous Supplementary information), we show the trend of $\Delta\sigma_{\text{WAL}}$ as a function of the conductivity σ_0 for various topological semimetals, where we plotted the data of eleven SrAs₃ crystals in total, including the results from the four additional samples (A1, A7-9). For the data from other topological semimetals, we plotted only the cases when the current jetting effect is carefully considered, following the reviewer's suggestion [A3-6]. In this revised plot, the $\Delta\sigma_{\text{WAL}}$ data of SrAs₃ form a cluster without showing a clear trend, which are somewhat smaller than those of other highly-conducting topological materials. As explained in our previous reply, Dirac or Weyl semimetals, in which the multiple band crossing points are located apart in the Brillouin zone, the size of $\Delta\sigma_{\text{WAL}}$ is very sensitive to the relative strength of large-angle (intervalley) scattering [Phys. Rev. B **92**, 035203 (2015)]. This is

because the small-angle (intravalley) scattering leads to the WAL due to π Berry phase of the backscattering trajectories, while the large-angle (intervalley) scattering without the associated Berry phase induces the competing weak localization (WL). Roughly speaking, the excess conductivity $\Delta\sigma_{\text{WAL}}$ due to the WAL is suppressed with the large-angle scattering. Because the large-angle scattering is also effective in reducing the semiclassical conductivity σ_0 , the measured $\Delta\sigma_{\text{WAL}}$ is expected to decrease in highly resistive samples with a small σ_0 . This captures a trend of $\Delta\sigma_{\text{WAL}}$ with variation of the conductivity σ_0 in topological semimetals as shown in the revised Fig. 4e.

We like to emphasize that such a rough argument on the overall trend of $\Delta\sigma_{\text{WAL}}$ for many topological semimetals cannot be applied to explain the detailed behaviors on a single system, here SrAs₃. In fact, our data from eleven samples in total are scattered in the plot, showing no clear trend. The detailed analysis of the $\Delta\sigma_{\text{WAL}}$ data only for SrAs₃ crystals reveals a clear trend of $\Delta\sigma_{\text{WAL}}$. As shown in the inset of Fig. 4e, the absolute magnitude of $\Delta\sigma_{\text{WAL}}$ varies systematically with the ratio between the radii of the nodal loop K_0 and the poloidal orbit κ . Such a systematic variation of $\Delta\sigma_{\text{WAL}}$ is consistent with the theoretical study on nodal-line semimetals [Phys. Rev. Lett. **122**, 196603 (2019)]. In nodal line semimetals, two quantum interference processes along the poloidal and toroidal directions compete with each other, determining the size of $\Delta\sigma_{\text{WAL}}$. As the radius of the poloidal orbit κ of the torus-shaped Fermi surface (FS) becomes smaller than the radius of the nodal loop K_0 ($\kappa \ll K_0$), the backward scattering trajectories associated with small-angle scatterings tend to encircle the π Berry flux in the poloidal plane, inducing dominant WAL effect. In contrast, as κ becomes close to K_0 , the scattering along the toroidal direction becomes sizable and introduces the WL contribution. Thus the absolute size of the observed $\Delta\sigma_{\text{WAL}}$ is expected to increase with the ratio K_0/κ , consistent with experiments. Together with 2D WAL behaviors of SrAs₃ as discussed in **A3-6**, these results support that WAL in the bulk SrAs₃ agrees well with the theoretically-predicted WAL behaviors for nodal-line semimetals.

In the revised manuscript, we removed the discussion on the relative size of $\Delta\sigma_{\text{WAL}}/\sigma_0$ of SrAs₃ in comparison with those of other topological semimetals. Instead, using our additional experiments and analysis, we emphasize that the unique characters of WAL of nodal line semimetals are clearly observed in our SrAs₃ crystals.

Q3-8. *One way to get information about the ratio of large angle scattering to small angle scattering is from comparing the mean free path from resistivity (only large angle scattering contributes) with the one from quantum oscillations (all scattering is contributing). What is this ratio in SrAs₃?*

A3-8. We appreciate the reviewer's helpful comments. The transport scattering time τ_t , estimated on the S2 is 0.42(5) ps, which is about 2.5 times bigger than average quantum scattering time $\tau_q \sim 0.17(3)$ ps, derived from the analysis on quantum oscillation data. The ratio between transport scattering time and quantum scattering time $\tau_t/\tau_q \sim 2.5$ is larger than the case of $\tau_t/\tau_q \sim 1$ with dominant large-angle scatterings. Together with two-dimensional weak antilocalization behavior [**A3-6**], these results indicate that the small-angle scatterings on the poloidal planes determine the weak antilocalization in SrAs₃.

Q3-9. *In line 129, the sentence is not finished.*

The number of significant digits is too large for both the effective masses and the scattering times, which usually have error bars of roughly 10 percent for the mass and larger for the scattering time.

A3-9. We thank the reviewer for careful reading on our manuscript. First, we have deleted unfinished sentence in the line 139. Also, we estimated the error bars of effective masses and scattering times, which are roughly 7% of their magnitudes, as the reviewer mentioned. In the revised manuscript, we presented the estimates with a proper error estimation.

List of the changes

1. Considering the reviewer #1 and #3's concerns about $\Delta\sigma_{\text{WAL}}/\sigma_0$ in **Q1-1** and **Q3-5**, we removed the discussion on the relative size of $\Delta\sigma_{\text{WAL}}/\sigma_0$ of SrAs₃ in comparison with those of other topological semimetals.

In addition, as the review #3 requested in **Q3-6**, we presented the dimensionality analysis on the WAL behavior in SrAs₃ together with the magnetotransport results of four additional SrAs₃ samples in the revised Fig.4 and Supplementary Note 7.

In the main text:

[Abstract, Line 31-32] "the quantum interference effect resulting in the two-dimensional behaviors of weak antilocalization."

[Introduction, Line 65-68] "are further corroborated by the quantum interference effect with disorder-induced scattering, resulting in unusual two-dimensional behaviours of weak antilocalization (WAL) and its strong variation to the FS characters."

[Results, Line 232-234] "What is unique for SrAs₃ is the unusual magnetic-field and temperature dependences of the magnetoconductivity $\Delta\sigma(H, T)$, that can be attributed to two main characters of the nodal-line fermions."

[Results, Line 249-270] "In this case, the dominant 2D WAL is expected to determine the magnetoconductivity ... These results strongly indicate the 2D nature of the WAL induced by nodal-line fermions in SrAs₃."

[Figure 4] New results on transverse magnetoconductivity ratio $\Delta\sigma(H)/\sigma(0)$ for additional four SrAs₃ samples, transverse magnetoconductivity $\Delta\sigma(H)$ with 2D and 3D WAL fittings, temperature-dependent phase coherence length l_ϕ and the excess conductivity $\Delta\sigma_{\text{WAL}}$ as a function of σ_0 for various topological semimetals with K_0/κ dependent $\Delta\sigma_{\text{WAL}}$ of SrAs₃ are presented.

In the Supplementary Information:

[Supplementary Note 7] New supplementary note explaining dimensionality of weak antilocalization in SrAs₃ is added.

[Supplementary Note 8] The discussion on the relative size of $\Delta\sigma_{\text{WAL}}/\sigma_0$ is removed.

[Supplementary Table S1] New results of additional four SrAs₃ samples (A1, A7-9) with error bars are added.

[Supplementary Fig. S4] New results on transport properties of additional four SrAs₃ samples (A1, A7-9) are added.

[Supplementary Fig. S10] New results on weak antilocalization of four SrAs₃ samples (A1, A7-9) are added.

[Supplementary Fig. S8] New results on dimensionality of weak antilocalization in SrAs₃ are presented.

2. Considering the reviewer #3 concerns (**Q3-1**), we state the limitation of the DFT-based calculations for the fine details of the low-energy band structures of SrAs₃ and emphasize good agreement between our model Hamiltonian and experimentally-determined band parameters.

In the main text:

[Results, Line 88-89] "the results of quantum oscillations, discussed below."

[Results, Line 156-158] "Moreover, the band overlap energy $\Delta \sim 120$ meV from our model calculations is consistent with that obtained by the optical conductivity measurements on our crystal."

3. As requested by the reviewer #3 in **Q3-3**, we added the detailed analysis on Zeeman splitting effect on quantum oscillations, together with the corresponding Landau fan diagrams in the revised Fig.S8 and the Supplementary Note 5.

In the Supplementary Information:

[Second and third paragraphs of Supplementary Note 5] “Before presenting the results from the detailed analysis, ... with spin splittings at different polar (θ) and azimuthal (φ) angles for the α and β orbits in Fig. S9”

[Supplementary Fig. S8] New results on Landau fan diagram with Zeeman splitting in SrAs₃ are presented.

4. Considering the reviewer #3 concerns in **Q3-5**, we toned down our statements on weak antilocalization and clearly stated that there are many remaining issues that requires further theoretical and experimental investigations in the revised manuscript.

[Discussion, Line 275-279] “There are several questions remained to be investigated, ... with the thinnest tubular FS and the largest K_0/κ among the NLSMs candidates and thus”

5. As requested by the reviewer #3 in **Q3-6**, we remove the data from the previous studies without careful consideration of current jetting effect in the revised Fig. S10i, Fig. 4e and Supplementary Table S3.

6. As the reviewer #3 suggested in **Q3-9**, we deleted unfinished sentence in the line 139 in the previous main text, and added properly estimated error bar in the revised main text and Supplementary Table S1, Table S2.

In the main text:

[Results, Line 134] “ $m^*/m_e = 0.076(5), 0.23(1),$ and $0.076(3)$ ”

[Results, Line 136] “ $\tau_q = 0.13(2), 0.12(2),$ and $0.015(1)$ ”

In the Supplementary Information:

[Supplementary Table S2] Properly estimated error bars of measured Frequency and effective mass are added.

7. Minor changes are listed below.

In the main text:

[References, Line 423] New reference added. “Jeon, J. W. & Choi, E. J. *Private communication.*”

[References, Line 436-437] New reference added. “Hikami, S., Larkin, A. I. & Nagaoka, Y. Spin-orbit interaction and magnetoresistance in the two dimensional random system. *Prog. Theor. Phys.* 63, 707–710 (1980).”

[References, Line 438-439] New reference added. “Lin, J. J. & Bird, J. P. Recent experimental studies of electron dephasing in metal and semiconductor mesoscopic structures. *J. Phys. Condens. Matter* 14, R501 (2002).”

In the Supplementary Information:

[Second paragraph of Supplementary Note 6] “As shown in Fig. S10i, we found a decreasing trend of $\Delta\sigma_{\text{WAL}}$ with lowering σ_0 .”

[References] New references added. “Wang, J et al. Vanishing quantum oscillations in Dirac semimetal ZrTe₅. *Proceedings of the National Academy of Sciences* 115, 9145–9150 (2018).”

[References] New references added. “Hikami, S., Larkin, A. I. & Nagaoka, Y. Spin-orbit interaction and magnetoresistance in the two dimensional random system. *Prog. Theor. Phys.* 63, 707–710 (1980).”

[References] New references added. “Lin, J. J. & Bird, J. P. Recent experimental studies of electron dephasing in metal and semiconductor mesoscopic structures. *J. Phys. Condens. Matter* 14, R501 (2002).”

REVIEWER COMMENTS

Reviewer #1 (Remarks to the Author):

All issues associated with my previous comments are addressed in the current version of manuscript by the authors.

Reviewer #3 (Remarks to the Author):

Kim et al. Nat. Com. July 2022

I have read the new version of the manuscript and the reply to the referee comments.

The authors have significantly improved the manuscript. They show that their analysis is careful and detailed and the article is well written.

I have some more comments. When they will be addressed, I will support the publication of this article in Nature Communications.

— Tilting of the bands and distortion of torus and its consequence on the quantum-oscillation phase

The authors now convincingly show that there is only a single-torus Fermi surface. They have detailed much better the information they have on the splitting of the bands Δ . However, they did not address E_{tilt} , which was part of my previous questions. The DFT calculation and Fermi surface presented in Fig. 1b and in Fig. S2 imply a sizeable tilt and hence, the torus is expected to have an irregular shape. Therefore, a splitting of the alpha frequency might indeed occur. Also, the croissant shape of the cross section of the torus should lead to a splitting of the beta frequency.

In order to rule out the influence of such expected splittings on the phase analysis and to further strengthen the claims, I suggest that the authors simulate a QO signal close to the experimental one, by allowing a split of the above frequencies. By doing then the quantum oscillation analysis exactly

analogue to the one done on the experimental data, they should check, which splitting and which phases would still be in agreement with the experimental data.

After the presentation of the electronic structure in the first section of the Results, where the tilted tori are presented, it is a bit surprising and misleading when in the next paragraph, an ideal torus is presented. This simplification might be valid (if above tests show that the allowed splitting is very small), but the authors should give a sentence that motivates or justifies this simplification.

— Line 167: shouldn't this be "a clear change from $-(0.3-0.4)$ to 0 near $\theta \approx 10^\circ$, when the poloidal orbit (α) is converted to the inner toroidal (β) orbit..." It would be helpful if the lower part of figure 3a would be expanded in y-direction so that the phases can be read off better in the graphs.

— Line 173: shouldn't this be "the same Φ_{SdH} as the inner orbit ..."

— Line 210: The Fermi energy shouldn't change. It is fixed in one material. Maybe the energy of the nodal line is changed or the dispersion changes. This and the following sentence should be reworded.

— I also have a comment to the answer A3-1. The discrepancy of the band overlap of DFT compared to experiment by optical spectroscopy is large. The authors call this "fine details of the DFT calculations", but 180 meV difference sounds rather large to me. In the TaAs family, difference of less than 10 meV were found.

— I highly appreciate the analysis done in A3-3. However, the empty red dots in the fan diagrams are hardly visible with that figure size. Also, it would be good to give the Φ_s from fits of the split Landau levels in the fan diagrams, and to give an example for at least one angle, how the numbers of Φ_s shown in figure 3A lower part were obtained.

Q3-1. Tilting of the bands and distortion of torus and its consequence on the quantum-oscillation phase

The authors now convincingly show that there is only a single-torus Fermi surface. They have detailed much better the information they have on the splitting of the bands Δ . However, they did not address E_{tilt} , which was part of my previous questions. The DFT calculation and Fermi surface presented in Fig. 1b and in Fig. S2 imply a sizeable tilt and hence, the torus is expected to have an irregular shape. Therefore, a splitting of the alpha frequency might indeed occur. Also, the croissant shape of the cross section of the torus should lead to a splitting of the beta frequency.

In order to rule out the influence of such expected splittings on the phase analysis and to further strengthen the claims, I suggest that the authors simulate a QO signal close to the experimental one, by allowing a split of the above frequencies. By doing then the quantum oscillation analysis exactly analogue to the one done on the experimental data, they should check, which splitting and which phases would still be in agreement with the experimental data.

A3-1. We highly appreciate the reviewer's helpful comments. As the reviewer pointed out, the schematic Fermi surface (FS) with a tilted nodal loop shown in the previous Fig. 1b is different from the shape of FS in SrAs₃ and can be misleading. In SrAs₃, the inversion symmetry of the crystal structure guarantees that the irregular deformation of the torus-shaped FS should be symmetric by inversion, as schematically presented in the revised Fig. 1b. Then for the magnetic fields within the poloidal plane, the sizes of the extremal orbits formed at the

FIG. 1. Crystal and electronic structures of a nodal line semimetal SrAs₃. **a**, The crystal structure of SrAs₃ **b**, The schematic band crossing for asymmetric nodal-line states with a tilted energy dispersion (E_{tilt}), a finite spin-orbit-coupling gap (Δ_{SOC}) and a band overlap energy (Δ). The corresponding Fermi surfaces at different Fermi levels (E_F) are shown in the right, a crescent-type for $E_{F,1}$ and a torus-type for $E_{F,2}$. **c**, The smoke-ring-type pseudospin texture imprinted on the Fermi surface. **d**, The Brillouin zone of SrAs₃ with a single nodal ring (red circle) centered at the Y point. **e**, The ARPES spectra of SrAs₃ taken at the Y point along k_x with the photon energy of 99 eV. The overlaid red and blue lines indicate the conduction and valence bands, respectively. **f**, The temperature dependence of the in-plane resistivity (ρ). The inset shows the carrier densities (n) for electron (e) and hole (h). **g**, The magnetic field-dependent Hall resistivity (ρ_{xy}) of SrAs₃ at different temperatures. **h**, A series of ARPES spectra taken along k_x at different photon energies (85-104 eV) corresponding to k_y , marked on top of each panel. **i**, The nodal-ring of the crossing points between the conduction and valence bands in ARPES data, with dashed red circle as a guide to the eye.

opposite sides of the torus-shaped FS in SrAs₃ should be the same. Therefore, the tiling of the nodal loop in SrAs₃ cannot induce splitting of the α orbit.

To further confirm the absence of the α -orbit splitting, we simulated SdH oscillations using the Lifshitz-Kosevich (LK) formula [Sov. Phys. JETP 2, 636-645 (1956)] and compared them with experiments, as the reviewer suggested. In Fig. R1b-R1c, we presented the best fit of the data for $H \parallel k_y$, using a single frequency F and a phase factor ϕ_0 as fitting parameters. We performed the LK fitting to the measured SdH oscillations, without (Fig. R1b) and with weighting on the high $1/H$ data (Fig. R1c). In both cases, the best fit reproduces nicely the experimental data, yielding the frequency $F \sim 12.7(2)$ T and the phase factor $\phi_0 \sim 0.117(6)$, consistent with the results from the analysis on the fast Fourier transform (FFT) spectra and the Landau fan diagram, shown in the main text. The LK fitting with two SdH frequencies were also carried out, but the best fit gives unrealistic values of a high Dingle temperature ($T_D > 100$ K) and a tiny frequency ($F < 1$ T) for the secondary oscillations. This additional contribution is found to be monotonous as a function of $1/H$ without any signature of the oscillatory behavior, which is most likely due to contamination of the oscillation data during background subtraction. The same conclusion can be obtained in the FFT analysis on the temperature dependent SdH oscillations for $H \parallel k_y$. As shown in Fig. R1a, a single dominant peak at $F \sim 12.9$ T is found in the FFT spectra, which shows systematic suppression with increasing temperatures. In contrast, a small shoulder near $F \sim 6$ T (the inset of Fig. R1a) exhibits no systematic trend with temperature, which cannot be a signature of additional SdH oscillations due to extra extremal orbits.

These observations strongly suggest that the possibility of the α orbit splitting by tiling of the nodal loop can be ruled out theoretically and experimentally.

For the β orbits, on the other hand, additional inner extremal orbits (β') can be formed above and below the nodal line plane due to the crescent shape of the poloidal cross-section for $H \parallel k_z$, as the reviewer pointed out (Fig. S8a). According to our model Hamiltonian, additional SdH oscillations with a frequency $\beta' \sim 18$ T is expected, as shown in the revised Fig. S8, which were not detected in experiments. In the FFT spectra of SdH oscillations for $H \parallel k_z$, a single dominant peak appears near $F \sim 32.5$ T with a small hump structure centered at $F \sim 0$ T (Fig. R1d). This hump features in the FFT spectra systematically weaken with lowering temperature, which is opposite to what is

FIG. R1. The fast Fourier transform (FFT) spectra and Lifshitz-Kosevich (LK) fitting to SdH oscillations of SrAs₃. **a, d**, The FFT spectra of SdH oscillations for $H \parallel k_y$ (**a**) and $H \parallel k_z$ (**d**) taken at different temperatures. The inset shows the enlarged view of the low frequency region. **b, c, e, f**, The LK fits (red solid lines) to the measured SdH oscillations (black solid circles) without (**b, e**) and with (**c, f**) weighting on the high $1/H$ data for $H \parallel k_y$ (**b, c**) and $H \parallel k_z$ (**e, f**).

expected for SdH oscillations. Accordingly, the LK fit to the measured SdH oscillation data, both unweighted or weighted for the high $1/H$ region, shows good agreement (Fig. R1e-R1f), yielding $F \sim 32.3(2)$ T and $\phi_0 \sim 0.01(2)$, consistent with the results from the Landau fan diagram shown in the main text. Thus, we conclude that the SdH oscillations with additional frequency β' are below the detection limit of our experiments.

The key parameters determining the amplitude of SdH oscillation are curvature factor $C = |\partial^2 A / \partial k_{\parallel}^2|^{-1/2}$ and cyclotron effective mass m^* , where A is the enclosed k -space area and k_{\parallel} is the k -component parallel to the magnetic field direction [Phys. Rev. Research **2**, 012055(R) (2020)]. From our model Hamiltonian of SrAs₃, we found that the β' orbit has a smaller curvature factor $C \sim 0.053$ and a larger cyclotron mass $m^* \sim 0.619m_e$ than those of the β orbit with $C \sim 0.103$ and $m^* \sim 0.246m_e$. Assuming that the additional β' orbit has Dingle temperature $T_D \sim 12$ K and g factor ~ 3 , similar to those of the β orbit obtained from SdH oscillations, we can estimate the expected oscillation amplitude of the β' orbit using the LK formula at $T = 1.8$ K and $H = 15$ T. The calculated amplitude for the β' orbit is ~ 100 times smaller than that of the β orbit, and such weak oscillations cannot be detected in our experimental conditions. Therefore, in the main text, we focused on the results of the β orbits. The discussion about possible oscillations for the additional β' orbits is provided in the revised Supplementary Note 4.

FIG. S8. Splitting of the inner toroidal orbits in SrAs₃. **a**, Two inner toroidal orbits, denoted by β and β' , in the torus Fermi surface with crescent-shaped cross-section, seen from different view angles. **b**, The expected SdH frequency (F) of the β' orbit from the model Hamiltonian discussed in the main text.

Q3-2. *After the presentation of the electronic structure in the first section of the Results, where the tilted tori are presented, it is a bit surprising and misleading when in the next paragraph, an ideal torus is presented. This simplification might be valid (if above tests show that the allowed splitting is very small), but the authors should give a sentence that motivates or justifies this simplification.*

A3-2. As schematically shown in Fig. 1b, distortion of the torus-shaped Fermi surface (FS) due to tilting of the nodal loop depends on the relative energy scales of tilting energy (E_{tilt}), band overlap energy (Δ), spin-orbit coupling energy (Δ_{SOC}), and Fermi level (E_F). According to our model Hamiltonian, constructed from the results of SdH oscillations, angle-resolved photoemission spectroscopy, and optical spectroscopy, the hierarchy of these energy scales is $\Delta \sim 120$ meV $>$ $E_F \sim 50$ meV $>$ $\Delta_{\text{SOC}} \sim 10$ -35 meV $>$ $E_{\text{tilt}} \sim 5$ meV. The tilting energy scale is the lowest, which results in small modification of the torus-shaped Fermi surface, including a weak azimuthal angle (ϕ) dependent SdH frequency, shown in Fig. 3a of the main text.

In the revised manuscript, we clearly state that the FS of SrAs₃ differs from the ideal torus-shaped FS, such as a crescent-shaped poloidal cross-section and a small distortion along the toroidal direction due to a small but finite tilting energy.

Q3-3. *Line 167: shouldn't this be "a clear change from $-(0.3-0.4)$ to 0 near theta approx 10 deg, when the poloidal orbit (alpha) is converted to the inner toroidal (beta) orbit..." It would be helpful if the lower part of figure 3a would be expanded in y-direction so that the phases can be read off better in the graphs.*

A3-3. We appreciate the reviewer’s careful reading on our manuscript. As the reviewer pointed out, the phase offset of SdH oscillation (ϕ_{SdH}) for the poloidal orbit (α) corresponds to $-(0.3-0.4)$, whereas ϕ_{SdH} for the toroidal orbit (β) corresponds to ~ 0 . We corrected the expression “a clear change from ~ 0 to $\sim 0.3-0.4$ near $\theta \sim 10^\circ$, ...” to “a clear change from $-(0.3-0.4)$ to 0 near $\theta \sim 10^\circ$, ...” in the revised manuscript. Also, we expanded the lower part of Fig. 3a to show a clear phase shift, following reviewer’s suggestion.

FIG. 3. Toroidal Fermi surface and Berry phase evolution of SrAs₃. **a**, Angle-dependent SdH frequency (F) and the phase offset of SdH oscillation (ϕ_{SdH}) for two samples S1 (black) and S2 (red). The spin-splitting phase (ϕ_s) and the characteristic phase (ϕ_0) are also shown in the lower panels. The calculated F using the model Hamiltonian is overlaid with red lines. The corresponding extremal orbits on the torus-shaped Fermi surface are also presented for selected field orientations in the inset. **b**, Torus-shaped Fermi surface of SrAs₃ with the poloidal orbit (α) and the inner (β) and outer (δ) toroidal orbits. **c**, Poloidal cross-section of the Fermi surface (α) with pseudospin textures indicated by the arrows. **d–g**, Landau fan diagram for various field orientations with different polar (θ) angles (**d,f**) and azimuthal (ϕ) angles (**e,g**) for S1. The maxima (solid circles) and minima (open circles) of $\Delta\rho(H)/\rho(0)$ are assigned as integer and half-integer of the Landau index. **h, i**, The second derivative of $\rho(H)$, $-d^2\rho/dH^2$, as a function of the normalized F/H for various magnetic field orientations with different polar (θ) (**h**) and azimuthal (ϕ) angles (**i**) for S2. The spin splitting peaks of SdH oscillations are indicated by triangle symbols. The shaded dashed lines correspond to the spin-split Landau levels, indicated by the color-coded integer index and the + and - symbols.

Q3-4. Line 173: shouldn’t this be “the same Φ_{SdH} as the inner orbit ...”

A3-4. In the revised manuscript, we corrected the expression “the same ϕ_{SdH} with the inner orbit ...” to “the same ϕ_{SdH} as the inner orbit ...” as the reviewer suggested.

Q3-5. Line 210: The Fermi energy shouldn’t change. It is fixed in one material. Maybe the energy of the nodal line is changed or the dispersion changes. This and the following sentenced should be reworded.

A3-5. We do appreciate the reviewer’s helpful comments. As reviewer pointed out, Fermi level is fixed in one material, but the energy levels of the band crossing points change in the case of the dispersive nodal lines. To make this point clear, we defined new energy parameter ε_F , the energy difference between the Fermi level E_F and the band crossing point in the momentum-energy space. When external magnetic field is rotated from $H \parallel k_y$ to $H \parallel k_x$, the SdH frequency decreases gradually, implying that the energy position of the Dirac node corresponding to the extremal poloidal orbit becomes closer to E_F , reducing $|\varepsilon_F|$.

Q3-6. I also have a comment to the answer A3-1. The discrepancy of the band overlap of DFT compared to

experiment by optical spectroscopy is large. The authors call this “fine details of the DFT calculations”, but 180 meV difference sounds rather large to me. In the TaAs family, difference of less than 10 meV were found.

A3-6. We highly appreciate the reviewer’s helpful comments. In general, the errors in the calculated band gap or band overlap energies can be as large as 10^2 – 10^3 meV, depending on the functional used for DFT calculations (Fig. R2) [Phys. Rev. Lett. **102**, 226401 (2009), J. Phys. Chem. Lett. **7**, 4165 (2016)]. For SrAs₃, we tested several different functionals for calculations and found that mBJ functional gives the best agreement with experiments with a moderate discrepancy of ~180 meV. This discrepancy is still sizable, but much smaller than that of ~500 meV obtained by GGA or LDA calculations (Supplementary Note 2 and Ref. 32). It is highly desirable to develop a better functional to produce the low energy band structures showing a more accurate agreement with experiments. However, we think, such a theoretical study is beyond the scope of this work.

FIG. R2. Calculated vs experimental band gaps for the various functionals, including mBJ functional (a) [Phys. Rev. Lett. **102**, 226401 (2009)] and other functionals (b) [J. Phys. Chem. Lett. **7**, 4165 (2016)].

Q3-7. I highly appreciate the analysis done in A3-3. However, the empty red dots in the fan diagrams are hardly visible with that figure size. Also, it would be good to give the Φ_s from fits of the split Landau levels in the fan diagrams, and to give an example for at least one angle, how the numbers of Φ_s shown in figure 3A lower part were obtained.

A3-7. We appreciate the reviewer’s helpful suggestion. Following the reviewer’s suggestion, we improved the visibility of Fig. S9, by making the spin-splitting data points (empty red dots) enlarged and removing the graphs without a clear spin-splitting features in the revised Fig. S9. In addition, we added vertical bars in the Landau fan diagrams to show the size of the splitting more clearly.

As we mentioned in the Supplementary Note 5, the spacing between two Zeeman-split peaks in the $1/H$ plot is ϕ_s/F and thus the spin-splitting phase ϕ_s can be estimated by multiplying the average spin-splitting spacing in the Landau fan diagram with the SdH frequency F . For example, under magnetic fields at $\theta = 8^\circ$ and $\phi = 90^\circ$ (Fig. S9d), the averaged Zeeman splitting spacing obtained for the 2nd and 3rd Landau levels is 0.0184 T^{-1} , and the SdH frequency is $F = 23.9 \text{ T}$, yielding the spin-splitting phase $\phi_s = 0.44$. Using the same method, we estimated the spin-splitting phase ϕ_s for $(\theta, \phi) = (13^\circ, 90^\circ)$ in Fig. S9e, $(\theta, \phi) = (90^\circ, 128.4^\circ)$ in Fig. S9f and $(\theta, \phi) = (90^\circ, 111.6^\circ)$ in Fig. S9g, which are $0.0244 \text{ T}^{-1} \times 22.5 \text{ T} = 0.55$, $0.0448 \text{ T}^{-1} \times 8.0 \text{ T} = 0.36$ and $0.060 \text{ T}^{-1} \times 9.7 \text{ T} = 0.58$, respectively. In the revised Supplementary Note 5, we added a paragraph describing how we extracted the spin splitting phase ϕ_s from the analysis of the Landau fan diagrams (Fig. S9).

FIG. S9. Landau fan diagram with Zeeman splitting in SrAs₃. **a-c**, The schematic illustrations of quantum oscillation peaks depending of the size of the spin splitting phase ϕ_s . The vertical bars under oscillation curves correspond to the Zeeman-split levels from the spin-degenerate Landau level. **d-g**, The quantum oscillations in the second derivative of resistivity, $d^2\rho/dH^2$, and the corresponding Landau fan diagram for S2. The vertical bars in the Landau fan diagram show the spin splitting spacing, which is expected to be field independent. The magnetic field directions are on the (k_y, k_z) plane (**d**, **e**) or (k_x, k_y) plane (**f**, **g**).

List of the changes

1. Considering the reviewer #3's concerns about additional quantum oscillation phase in **Q3-1**, we removed schematic Fermi surface of nodal-line semimetals without inversion symmetry in Fig. 1b. Instead we presented a characteristic Fermi surface of nodal-line semimetals with inversion symmetry like SrAs₃ in the revised Fig. 1b. Moreover, in the revised Supplementary Note 4 and Fig. S8, we provided detailed discussion for the additional toroidal orbit β' .

In the main text:

[Figure 1] The revised schematic Fermi surfaces with inversion symmetry is presented (Fig. 1b).

In the Supplementary Information:

[Third paragraph of Supplementary Note 4] "For the torus FS with crescent-shaped cross-section, additional inner extremal orbits (β') also can be formed above and below the nodal line plane ... we focus on the β orbit in the main text."

[Supplementary Fig. S8] New results on additional β' orbit in SrAs₃ are presented.

2. As the reviewer #3 suggested in **Q3-2**, we clearly state that the Fermi surface of SrAs₃ differs from the ideal torus-shaped Fermi surface and has small distortion along the toroidal direction due to a small but finite tilting energy.

In the main text:

[Results, Line 151-154] "Along the toroidal direction, a finite tilting energy $\Delta_{\text{tilt}} \sim 5$ meV, ... leading to a weak variation of the SdH frequency."

3. As requested by the reviewer #3 in **Q3-3**, we corrected the statement on the offset of SdH oscillations ϕ_{SdH} and enlarged the lower part of Fig. 3a to improve visibility in the revised manuscript.

In the main text:

[Results, Line 171] "-(0.3-0.4) to 0"

[Figure 3] The lower part of Fig. 3a is expanded.

4. As the reviewer #3 suggested in **Q3-4**, we corrected preposition in the revised main text.

In the main text:

[Results, Line 177] "as"

5. Considering the reviewer #3's concerns in **Q3-5**, we clearly defined new energy parameter ε_F to avoid misleading.

In the main text:

[Results, Line 96-97] " ε_F , the energy difference between E_F and the band crossing point in the momentum-energy space,"

[Results, Line 210-215] "Thus, the ϕ -dependence in both the SOC gap (Δ_{SOC}) and the ε_F introduces ... as the magnetic field approaches to $H \parallel k_x$."

6. As requested by the reviewer #3 in **Q3-7**, we enlarged the graphs for Landau fan diagrams and also the spin-splitting data points (empty red dots) in the revised Fig. S9. In addition, we explained how we extracted the spin splitting phase ϕ_s , and added representative examples showing a sizable spacing between the spin-split Landau levels in the revised Supplementary Note 5 and Fig. S9.

In the Supplementary Information:

[Supplementary Fig. S9] Figures without spin-splitting features are removed, and Landau fan diagrams and red dots are enlarged. Vertical bars showing splitting spacing are also added.

[Second paragraph of Supplementary Note 5] “We note that the spin-splitting appears to be constant ... from the average spin-splitting spacing multiplied by the SdH frequency F ”

7. Minor changes are listed below.

In the Supplementary Information:

[References] New references added. “Shoenberg, D. *Magnetic Oscillations in Metals* (Cambridge Univ. Press, Cambridge, 1984).”

[References] New references added. “Lifshitz, I. M. & Kosevich, A. M. Theory of magnetic susceptibility in metals at low temperatures. *Sov. Phys. JETP* **2**, 636–645 (1956)”

REVIEWERS' COMMENTS

Reviewer #3 (Remarks to the Author):

The authors have replied in detail and with much care to all of my remaining concerns. In particular, they have corrected some fundamental symmetry-related issues with the shown Fermi surfaces and ruled out other possible analysis of their data. I think the article is now much clearer and profound than in the beginning.

With all the input I gave, I should be one of the coauthors.